# Modelling and measuring complexity of traditional and ancient technologies using Petri nets

Sebastian Fajardo[1]☺*, Jetty Kleijn[2]☺, Frank W. Takes[2]☺, Geeske H. J. Langejans[1,3]☺*

1 Department of Materials Science and Engineering, Delft University of Technology, Delft, The Netherlands,
2 Leiden Institute of Advanced Computer Science (LIACS), Leiden University, Leiden, The Netherlands,
3 Palaeo-Research Institute, University of Johannesburg, Johannesburg, South Africa

☺ These authors contributed equally to this work.
* s.d.fajardobernal@tudelft.nl (SF); g.langejans@tudelft.nl (GHJL)

**Data Availability Statement:** All relevant data are within the paper and its Supporting Information files.

**Funding:** This research is part of the ancient adhesives project, funded by the European

## Abstract

Technologies and their production systems are used by archaeologists and anthropologists to study complexity of socio-technical systems. However, there are several issues that hamper agreement about what constitutes complexity and how we can systematically compare the complexity of production systems. In this work, we propose a novel approach to assess the behavioural and structural complexity of production systems using Petri nets. Petri nets are well-known formal models commonly used in, for example, biological and business process modelling, as well as software engineering. The use of Petri nets overcomes several obstacles of current approaches in archaeology and anthropology, such as the incompatibility of the intrinsic sequential logic of the available methods with inherently non-sequential processes, and the inability to explicitly model activities and resources separately. We test the proposed Petri net modelling approach on two traditional production systems of adhesives made by Ju/'hoan makers from Nyae, Namibia from *Ammocharis coranica* and *Ozoroa schinzii* plants. We run simulations in which we assess the complexity of these two adhesive production systems in detail and show how Petri net dynamics reveal the structural and behavioural complexity of different production scenarios. We show that concurrency may be prevalent in the production system of adhesive technologies and discuss how changes in location during the process may serve to control the behavioural complexity of a production system. The approach presented in this paper paves the way for future systematic visualization, analysis, and comparison of ancient production systems, accounting for the inherent complex, concurrent, and action/resource-oriented aspects of such processes.

## 1. Introduction

Recently, the complexity of ancient and traditional production systems has gained interest [1–3]. These production systems are considered informative about the evolution of technology and the human mind [4, 5]. They also provide insight into the transmission and maintenance

Research Council (ERC) under the European Union's Horizon 2020 research and innovation programme grant agreement No. 804151 (grant holder G.H.J.L). The funders had no role in study design, data collection and analysis, decision to publish, or preparation of the manuscript.

**Competing interests:** The authors have declared that no competing interests exist.

of knowledge and the societies behind those systems [6–9]. Archaeologists and anthropologists use the *chaîne opératoire* method and other approaches to model and analyse production systems [10–15]; and as underlying principle for current methods to assess complexity [16–19]. However, there are several issues with these approaches that hamper agreements about what constitutes complexity and how we can compare the complexity of production systems. We consider that computational modelling of production systems can help resolve these issues and will provide a replicable method to analyse complexity. Other disciplines, such as software engineering, business process modelling, and computer science face similar challenges while studying complexity, and an entire field of process modelling research emerged to tackle these problems [20–22]. There, Petri nets [23, 24] are common models used to effectively model and assess system complexity. In this paper we propose an approach to use Petri nets to study the behavioural and structural complexity of production systems.

In the following pages we present the advantages of Petri nets to model and measure the complexity of technological systems. After introducing Petri nets, we test this approach by modelling two adhesive production systems. Adhesives are made by Ju/'hoan makers from Nyae, Namibia from *Ammocharis coranica* (Ker Gawl.) Herb. (Amaryllidaceae) and from *Ozoroa schinzii* (Engl.) R. Fern & A. Fern (Anacardiaceae) plants [25]. Finally, we study the complexity of these two adhesive production systems and show how Petri nets provide a formal way to study production systems, revealing the structural and behavioural complexity of different production scenarios.

## 1.1. Petri nets: Non-sequential formal models

We consider Petri nets a well-established approach to model production systems and study their technological complexity. Petri nets emerged in the 1960's as a counterpart to sequential models like state machines [26, 27]. Over the years, they developed into a vast framework to design and study distributed systems consisting of concurrently operating agents, that is components that operate independently except for occasional exchanges of messages or resources or to synchronize certain activities [cf. 23, 24]. This has led to a wide range of methods and automated techniques in, for example, structure-based analysis, verification and model checking, and system synthesis [e.g. 28–31]. Examples of successful applications of Petri nets include, hardware design, biochemical networks, business process modelling, and manufacturing systems [e.g. 21, 32–38].

Petri nets are a flexible and robust framework comprising many different families of nets, ranging from fundamental classes like Elementary Net Systems [cf. 39] to high level models like Coloured Petri Nets [cf. 40]. Basic Petri nets are easily extended with features to facilitate an explicit representation of quantities of resources as in Place/Transition Systems [cf. 41], different types of resources (as in Coloured Petri Nets), and aspects related to time and stochastics [42, 43]. An advantage of the model is that it allows one to both specify and design the concurrent behaviour of distributed systems [44]. Petri net models are used to analyse and compare systems using various metrics and methods to assess, among others, system performance, probability of events, and behavioural and structural complexity [e.g. 45–50]. Moreover, new metrics and methods can be developed using the underlying mathematics of Petri nets.

Modelling with Petri nets has advantages in the analysis of complexity. First, Petri nets are based on local interactions which determine concurrency, conflict, and causality relations within a system. With Petri nets it is possible to represent a system's states in a distributed way and to model its actions purely locally involving only those parts of the system that are directly affected. So rather than time, structure and local states determine the relations between events and resources. In current approaches, the progress of time defines the relations between events

[12 p. 106, 51 p. 253, 52 p. 31], neglecting the effects of concurrency and asynchronous events in the complexity of a production system. Models generated with this time-based principle impose an order of events that strictly speaking is not enforced by the process. For example, sometimes hunter-gatherer groups prefer to preserve fire than to create it anew [53]. In these cases fire is available before the procurement of raw materials or the start of any other production event. Modelling techniques able to represent asynchronous events can grasp properly the effects of such behaviours.

An additional advantage of Petri nets is that local states and local state changes are determined as part of the modelling process. Entities relevant to the system and their interactions can be represented in the structure and hence in the local states. Previous studies show that to understand technical processes, resources, events, and their relations should be systematically distinguished in the model, inside and between stages of the production process [2]. When modeling a production process as a Petri net, it is possible to describe explicitly the raw materials, the production of tools, and the people involved in the process. This allows one to study the intricate relations between the actors, the resources and the events that may take place. Here we stress that Petri nets are not a static model. On the contrary, the dynamic (behavioral) aspect is crucial, and a Petri net may have many (concurrent) runs that all start from the same initial state. Current approaches use arbitrary stages and possibly arbitrarily defined boundaries to aggregate events in the models and mark system's states as global changes [12, 54]. Using these stages to define the states of the system hides the causal relations between entities and the effects of variables such as the preferences about products, the number of individuals involved, or the availability of raw materials. For example, preferences involved in a production system, may alter causal relations between technological behaviours [55]. These preferences and other variables may change the state of the system as a whole or some parts of the system at a given moment, making it less or more demanding to obtain a product.

An important advantage of Petri nets is that they are a graphical model with a clear intuitive understanding. This makes it possible for people unfamiliar with the formalities, to grasp the structure and develop insight into the dynamics of the modeled system. Without a systematic, intuitively, and consistent representation, the boundaries of the modelled systems are easily ignored. Current approaches present production systems using natural language [12, 19, 56–58], matrices and networks [18, 59], cognigrams [16, 17, 60], or a combination of these [10, 61–63]. The variability in representation is generated in part by the abundance of production systems among societies. However, most variability stems from the difficulties to formalize and connect structural (static) and behavioural (dynamic) system aspects.

Finally, Petri nets provide a formal modeling tool with a solid mathematical foundation. This has led, as mentioned earlier, to an extensive tool kit to investigate and compare Petri nets with respect to relevant and possibly newly defined concepts of complexity. Using relatively informal concepts often requires to measure complexity focusing on one aspect of the system. For example, the number of problems to solve [16] or the number of steps in the process [19]. Without formal definitions for system's elements and relations, few concrete quantitative assessments can be done regarding the system structure implications on behaviour and about the dynamics of systems.

We argue that the characteristics mentioned above make Petri nets a promising theoretical and methodological addition to current research of the degree of complexity of technological systems. To show this, we model the makers, actions, tools, and materials of two adhesive production processes as Petri nets. More specifically, we implement the models as Place/Transition nets [41]. We use Snoopy software [64] to visualize the structure and dynamics of the models. We compute reachability graphs with TINA toolbox [65] for multiple scenarios of the models with different number of makers to analyse the behaviour and complexity of the processes.

## 2. Methodology

In this section we introduce Place/Transition (P/T) nets [41]. They are the most well-known family of Petri nets, often considered as the archetypal Petri net model underlying many higher level net models. Being based on natural numbers they are more appealing from a modelling point of view than Boolean nets like the Elementary Net Systems. From now on, we will refer to P/T nets simply as Petri nets.

### 2.1. Definitions

Petri nets are structured as directed bipartite graphs, that is, they have two types of nodes (places and transitions) with arcs between nodes of different type. Places represent passive information (e.g., resources or conditions) and transitions represent active elements, the occurrence of which is determined by and affects the information of the places that are adjacent to them. This adjacency is determined by the arcs. Moreover, the arcs have weights. Graphically (cf. Fig 1), places are drawn as circles. transitions as rectangles, and arcs as arrows with their weight as a label (if it is two or more; weight one is not depicted).

**Definition 1.** A *Petri net* is a tuple ($P$, $T$, $F$, $W$) where $P$ is a finite nonempty set of *places*, $T$ is a finite nonempty set of *transitions* such that $P$ and $T$ are disjoint (have no elements in common),

$F \subseteq (P \times T) \bigcup (T \times P)$ is the flow relation defining arcs from $P$ *to* $T$ and from $T$ to $P$, and $W$: $F \to \mathbb{N}^+$ is a function associating a positive integer as arc weight to each arc in $F$.

For technical convenience, we extend the weight function to a function from $(P \times T) \bigcup (T \times P)$ to $\mathbb{N}$ (all nonnegative integers) by setting $W(x, y) = 0$ if and only if $(x, y)$ is not an arc in $F$.

The dynamics of a Petri net is based on the concept of states (often called *markings)* and state changes.

**Definition 2.** A marking of a Petri net ($P$, $T$, $F$, $W$) is a function $M$: $P \to \mathbb{N}$.

Thus a marking associates a nonnegative integer with each place of the Petri net. Graphically, a marking $M$ is represented by drawing in each place $p$ its associated number $M(p)$ of *tokens* (black dots, see Fig 1). When a place represents a resource, the number of tokens in that place is a quantification of the availability of that resource in the current state (marking). States are changed through the occurrence of transitions as defined next.

**Definition 3.** Let $N = (P, T, F, W)$ be a Petri net. Let $t$ be a transition of $N$.1.

1. Let $p$ be a place of $N$. Then $p$ is an input place of $t$ if $(p, t)$ is an arc in $F$; similarly, $p$ is an output place of $t$ if $(t, p)$ is in $F$.2.

2. Let $M$ be a marking of $N$. Then $M$ enables $t$ (to occur) if $M(p) \geq W(p, t)$ for each input place of $t$.3.

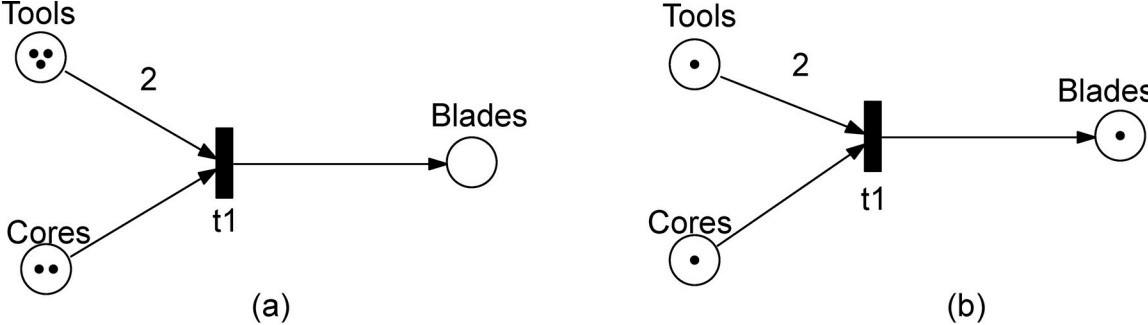

**Fig 1. Blade production model to illustrate the graphical representation, markings and firing rule of Petri nets.**

3. The occurrence of $t$ leads from $M$ to a marking $M'$, denoted $MttM'$, if $M$ enables $t$, and $M'$ is defined by $M'(p) = M(p) — W(p, t) + W(t, p)$ for all places $p$.

Thus, to be enabled, a transition needs at least as many tokens in each of its input places as the weight of the corresponding arc indicates. When the transition occurs, it 'consumes' these tokens and 'produces' in each of its output places the number of tokens indicated by the arc weights. Note that places not adjacent to a transition play no role in its enabling and are not affected when it occurs. Consequently, the occurrence of a transition is by local enabling and has a local effect. Often the occurrence of transitions is referred to as '*firing*'. Points 2 and 3 in Definition 3 together constitute the *firing rule* of Petri nets. The firing of a transition is instantaneous and the choice of which transition to fire when several transitions are enabled at the same time is random. When several transitions are enabled by a marking with sufficient tokens to fulfil the input requirements of each transition, then these transitions may occur concurrently at that marking.

**Definition 4.** A subset $T' \subseteq T$ is concurrently enabled at marking $M$ if and only if for all $p \in P$ it is true that $\sum \{W(p,t): t \in T'\} \leq M(p)$.

Fig 1 shows in a small Petri net a fragment of a blade production process [66]. Two tools, a percussor and a punch, are used to extract a blade from a core platform. The Petri net has a place 'Tools' to represent the available tools, a place 'Cores' for cores, and a third place 'Blades' for blades. Initially, three tools and two cores are available and there are no blades, represented by the marking of the places in Fig 1A, which can also be written as marking $M_0 = (3,2,0)$. Thus transition t1 is enabled in this marking as, according to the arc weights, it needs two tools and one core to occur. If it fires, we have a new marking $M_1 = (1, 1, 1)$, as depicted in Fig 1B with one tool and one core left and one blade produced.

## 2.2. Reachability graphs

One way to assess the complexity of a production system is by identifying its number of reachable states (markings). We argue that systems with low behavioural complexity will show small state spaces, and high behavioural complexity is represented by large state spaces. More reachable states are an indication of more potential for variation in the events that may occur in the evolution of the system. Makers may have to process more information to get from one state to another or to reach a final state. Actions that may occur concurrently increase the number of reachable states. Also, when a resource in the system controls the enabling of concurrent actions, changing that resource shows when and how the potential for concurrency is maximized. In a Petri net that starts from an initial marking, the firing rule determines the markings reachable from the initial state and the paths leading to them. In case an initially marked Petri net has a finite number of reachable markings, this leads to a finite *reachability graph*. If there are infinitely many markings reachable from the initial marking, a *coverability graph* can be used as a finite representation [e.g. 41].

In the reachability graph of a Petri net, each node is a reachable marking thus represents a possible state of the modelled system. All these possible states together form the state space of the system. The arcs between two nodes represent the occurrence of a transition leading from the first marking to the second. Reachability graphs are initialized with the initial marking of the Petri net.

**Definition 5.** Let $N = (P, T, F, W)$ be a Petri net and let $M$ be a marking of $N$.

1. $(N, M)$ is called a *marked Petri net*.

2. The set of markings reachable from $M$ in $N$, denoted by $R(N, M)$ is the smallest set such that $M$ is in $R(N, M)$ and whenever $M_1$ is in $R(N, M)$ and marking $M_2$ is such that $M_1ttM_2$ for some transition $t$ in $T$, then $M_2$ is also in $R(N, M)$.

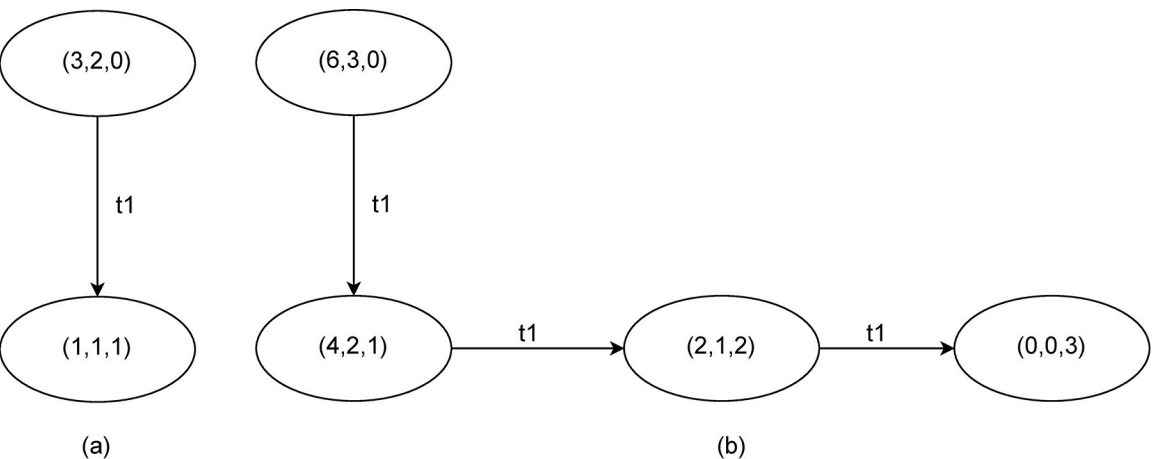

**Fig 2. Two reachability graphs for the blade production model.** (a) Reachability graph for the blade production model with an initial marking with two tools and one core available. (b) Reachability graph for the blade production model with six tools and three cores available. Ovals represent states with the corresponding marking written inside. Directed arrows represent edges and correspond to an occurrence of transition t1.

3. The *reachability graph of* $(N, M)$ is the initialised, arc labelled, directed graph $RG(N, M) = (R(N, M), E, M))$ with set of nodes $R(N, M)$, initial node $M$, and set of edges $E \subseteq \{(M_1, t, M_2): M_1, M_2$ in $R(N, M)$ and $M_1ttM_2\}$.

Fig 2 shows two reachability graphs for the blade production model (Fig 1) with two different quantities of tools and cores available in the initial state of the system. The graph in Fig 2A shows that two markings are reachable when initially three tools and one core are available. Fig 2B shows the reachability graph for an initial marking with six tools and three cores.

Below, we switch from the sequential descriptions of a *chaîne opératoire* approach to Petri nets by focusing on the dynamics of elements that interact in production systems and the events that determine these interactions. We use *chaîne opératoire* descriptions in natural language of real adhesive production systems as the input for the models. We identify the goal of the models and the abstraction level, including the assumptions used to build the models, the active and passive elements in the system, the information carried by tokens, the rules of scheduling of the events, and the auxiliary elements to facilitate the communication of the elements in the system.

## 3. Results. Two Ju/'hoan adhesive productions

To demonstrate the analytical advantage of Petri nets, we constructed Petri nets for the ethnographic descriptions of the production of two traditional adhesive materials made by Ju/'hoan makers in Nyae, Namibia [25]. The Ju/'hoansi maintain traditional knowledge about hunting practices and plant gathering. Wadley and colleagues documented their production of adhesives materials, poison, and arrow hafting. Ju/'hoan makers produce at least three different adhesives made from three different plants: *A. coranica*, *Terminalia sericea* Burch. ex DC. (Combretaceae), and *O. schinzii*. *A. coranica* adhesive is used to haft heavy duty tools, such as spears and axes, and to repair other objects. *T. sericea* and *O. schinzii* adhesive are used to haft and poison arrowheads. Here we modelled the procurement of raw material and production of adhesives made of *A. coranica* and *O. schinzii*. For both production processes we dedicate a subsection to I) describing the processes and II) explaining our modelling assumptions, followed by two subsections for each of the two processes, providing the resulting Petri net

models. Finally, section 3.4 presents simulations of the reachability graphs that effectively assess the complexity of each process.

## 3.1. Petri nets models for two Ju/'hoan adhesives

**I. Description of the processes.** Here we summarize the *chaîne opératoire* descriptions for the *A. coranica* and *O. schinzii* adhesive production that can be found in [25]. Ju/'hoan makers use digging sticks, fire sticks, and knives from their personal toolkits to make *A. coranica* adhesive. During the procurement of raw materials, the makers collect *A. coranica* bulbs, *Dichrostachys cinerea* (L.) Wight & Arn. (Fabaceae) (sickle bush) branches, grass, twigs, small branches, and calcrete blocks. The description of the *A. coranica* adhesive includes at least eight subprocesses. (1) The makers search for *A. coranica* bulbs and dig them with digging sticks from the sand. (2) Sickle bush branches and firewood composed by grass, twigs, and other small branches are collected from the surface at the location of *A. coranica* plants. (3) The makers light a fire using firesticks and the collected firewood. They use sickle bush branches to make charcoal for heating the scales. (4) Calcrete blocks are also collected from the surface surrounding the *A. coranica* plants. (5) They peel the bulbs with a knife to use the inner scales. The outer scales and other plant parts are discarded. (6) The makers extract, stack, and dust the scales from the bulbs. Charcoals are arranged on one side of the fire and the scales are heated on both sides on top of the charcoals, initially one at the time, later few at a time. Most of the ash and charcoal on the scales is flicked off, leaving only small fragments. (7) The heated scales are pounded using large and small calcrete blocks as anvils and hammer stones, respectively. During pauses the scales are folded inward until they turn into a soft pulp, which is (8) kneaded by hand and piled together in a ball. During the pounding and kneading, new scales are placed on charcoals and monitored to prevent burning. The process ends by reheating the pulp ball briefly and kneading it one more time to make a glue ball.

The production of *O. schinzii* adhesive requires from the toolkit of Ju/'hoan makers digging sticks, fire sticks, knives, and one glue carrier that serves as storage device. The makers collect the following raw materials: *O. schinzii* roots, branches from *T. sericea* and *Combretum* sp. (Combretaceae), a *Grewia flava* DC. (Malvaceae) branch, and *Aristida adscensionis* L. (Poaceae) grass. At least eight subprocess are suggested in the ethnographic description. (1) The makers expose the roots of a group of *O. schinzii* bushes by excavating sand around them using their digging sticks. Roots are cut with their knives and the holes around the *O. schinzii* bushes are filled again with sand to prevent death of remaining plants. (2) *T. sericea* and *Combretum* sp. are collected for firewood. The roots and firewood are collected in one location and then makers move to process materials in their homes. (3) Fire and coals are produced using fire sticks and *T. sericea* and *Combretum* sp. branches to heat the sliced roots. (4) The makers carve an applicator from the *G. flava* branch to extract the latex that exudates from the heated roots. (5) The makers cut slits on the amputated roots using knives. (6) *A. adscensionis* grass is lit using the fire and burnt completely by lifting it with a stick. The resulting ashes are crushed by hand to form a fine black powder. (7) Roots are heated to let latex exudate. (8) They use the applicator to transfer the latex to the glue carrier. The process ends when several layers of the black powder are pressed into the surface of the latex ball on the glue stick.

**II. Abstraction level and modelling decisions.** The Petri nets model the procurement of raw material and production of adhesives under the assumption of a static environment providing sufficient resources. We thus can focus on the intrinsic invariance of the glue making process without concern for how occasional environmental circumstances might influence the production steps.

We also do not distinguish between individual makers and individual resources. The number of makers is not a priori fixed but given as a parameter of the model. Our Petri nets model

people executing actions and processing resources. Expert knowledge and decisions are not represented. We use transitions to represent actions and places to represent resources. Resources include raw materials, subproducts, tools, and makers.

Markings (the number of tokens in each place) represent the availability of resources with each token representing a logical minimum unit. Note that some places and transitions are used to control the flow of the process and may be used, for example, to check that certain threshold values have been reached. Arc weights represent the number of resources required as input for an action or the number of items produced.

### 3.2. A Petri net model for *A*. *coranica* adhesive production

From the *chaîne opératoire* description it follows that the *A*. *coranica* adhesive production can be seen as consisting of activities and resources that generate inputs such as firewood or bulb scales that are required in the process, or the final adhesive product, which in this case is a glue ball. All subnets have *p* people available with *ds* digging sticks, *k* knives, and *fs* firesticks, usually one per person, to represent the toolkit of each maker. The subprocesses that group the activities are listed below following the order of the ethnographic description summarized in section 3.1:

1. Dig Bulbs,

2. Collect Firewood and Branches,

3. Light Fire,

4. Collect Small and Large Calcrete Blocks,

5. Prepare Bulbs,

6. Heat Selected Scales,

7. Pound,

8. Knead.

By identifying the common resources, the input/output flow, and the links between steps, these subprocesses can be taken together to form the Petri net model as whole. They are not necessarily ordered, but some may need input from others. Subprocess 3, for example, requires firewood and branches that are collected in subprocess 2. Below we discuss the Petri net models for each of these subprocesses. The subprocesses are depicted in Figs 3–10. Note that places and transitions that connect different submodels are shaded grey and have the same name.

*Subprocess 1 Dig bulbs* (Fig 3). The net for this subprocess has a place 'Start 1' initially marked with one token, to guarantee that the process is executed at most once. The place '# Collected bulbs' controls the end of the subprocess by counting whether the required *nb* bulbs were dug. Here *d* people can take part in the digging using *ds* digging sticks. Note that subprocess 5 can begin only after subprocess 1 is finished.

*Subprocess 2 Collect Firewood and Branches* (Fig 4). This subprocess collects grass, twigs, and small branches, modelled by places 'Firewood' and 'Branches'. The subprocess has start place 'Start2' that is initially marked. We assume that the process stops when *nfw* units of firewood and *nbr* units of branches have been collected, which is controlled by the places '# Collected firewood' and '# Collected branches'. Note that collecting firewood and collecting branches may interleave and could even happen concurrently if more than one person ($c \geq 2$) is involved. Subprocess 2 should have been completed before subprocess 3 can start.

*Subprocess 3 Light Fire* (Fig 5). This subprocess uses the *nfw* firewood and *nbr* branches collected in subprocess 2. Here a fire is ignited and all *nbr* branches are added. The subprocess

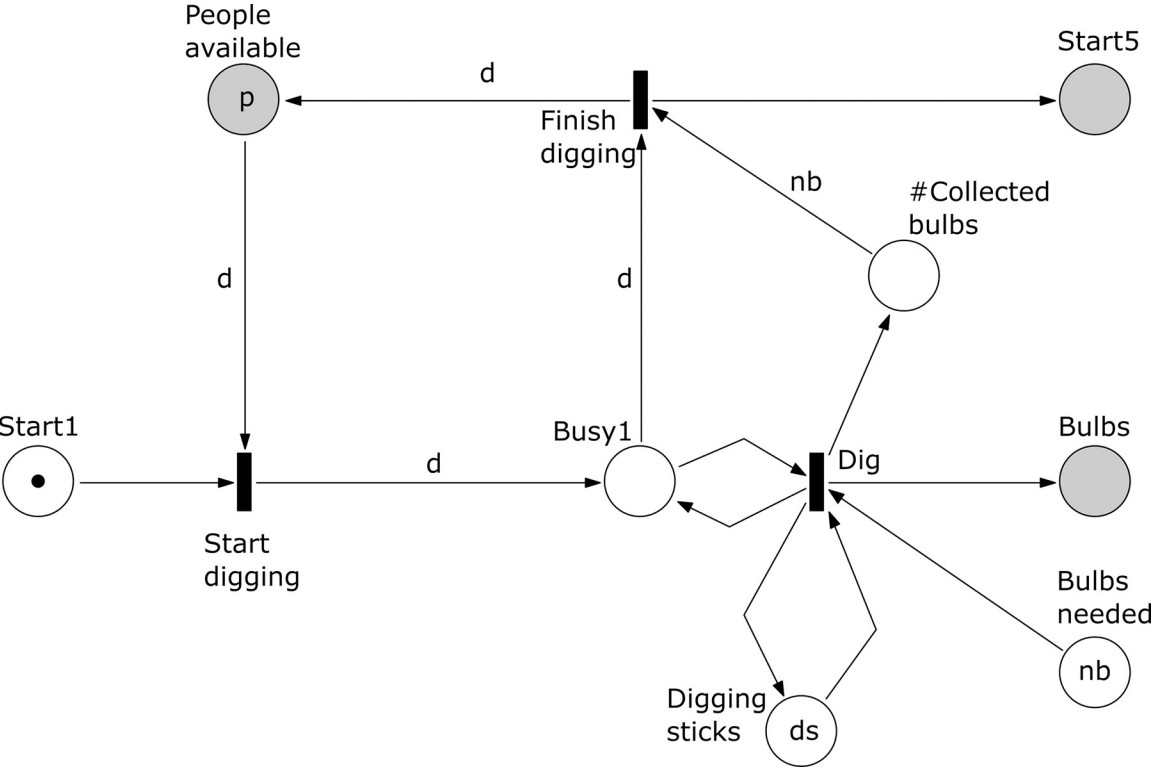

**Fig 3. Subprocess 1 Dig Bulbs of the *A. coranica* model.**

requires one person that can start the fire and add branches but note that due to place 'Fire' the transition 'Add branches' can occur only after the transition 'Light fire' has occurred. This subprocess produces fire and coals represented by a single token in the place 'Fire and coals'.

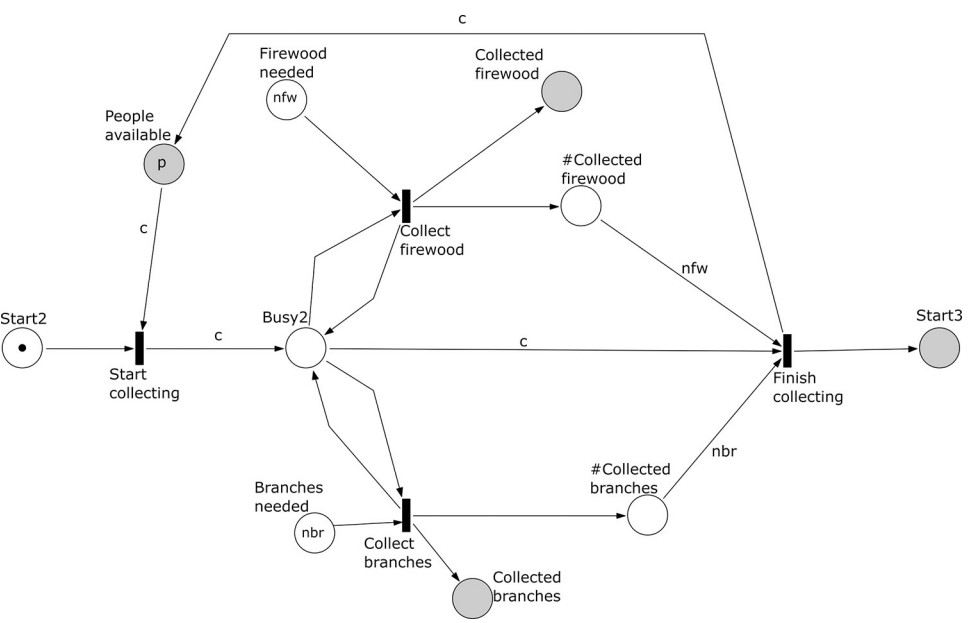

**Fig 4. Subprocess 2 Collect Firewood And Branches of the *A. coranica* model.**

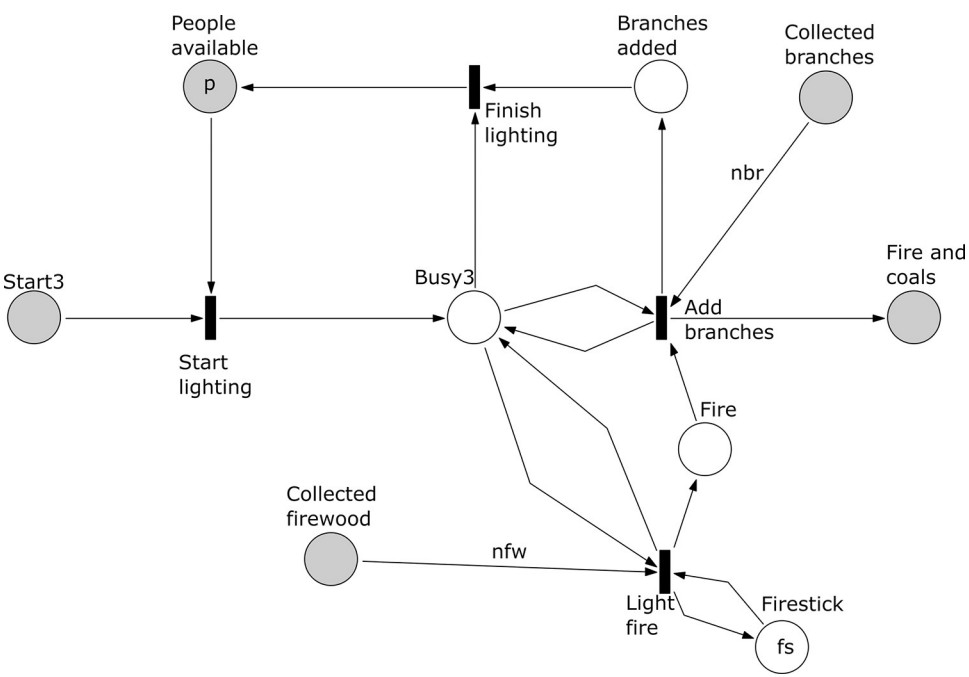

**Fig 5. Subprocess 3 Light Fire of the *A. coranica* model.**

*Subprocess 4 Collect Small and Large Calcrete Blocks* (Fig 6). This subprocess is initially marked with a token in the place 'Start4' and does not require any tool to be executed. Here *nbs* small blocks and *nbl* large blocks are collected. The large ones are used as anvils and the small ones serve as hammer stones. The number of people involved is *b* and they can either look for large or small blocks. If there are at least two people involved ($b{\geq}2$) searching for two

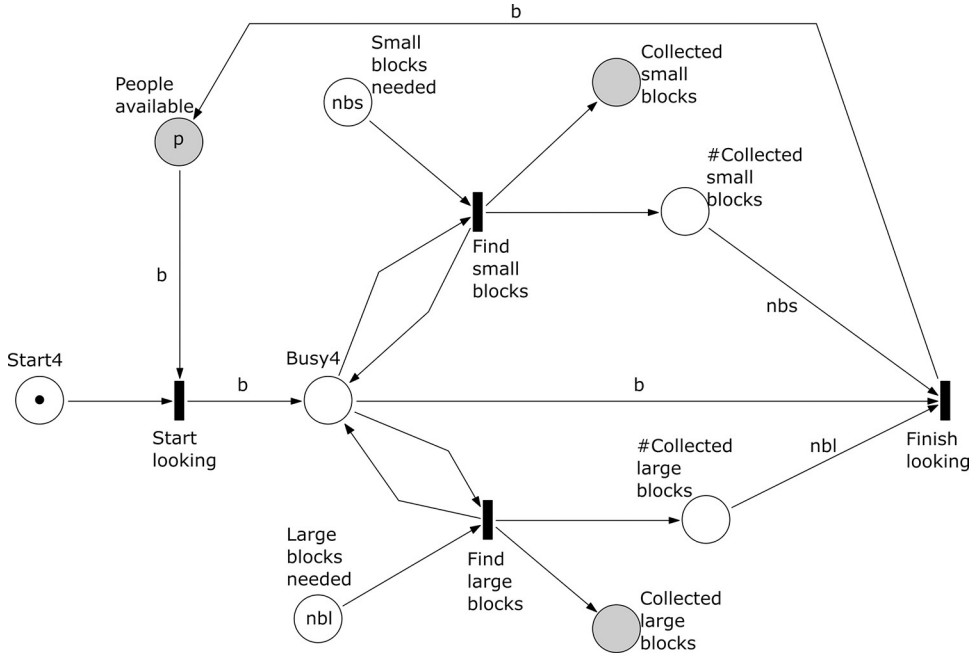

**Fig 6. Subprocess 4 Collect Small and Large Calcrete Blocks of the *A. coranica* model.**

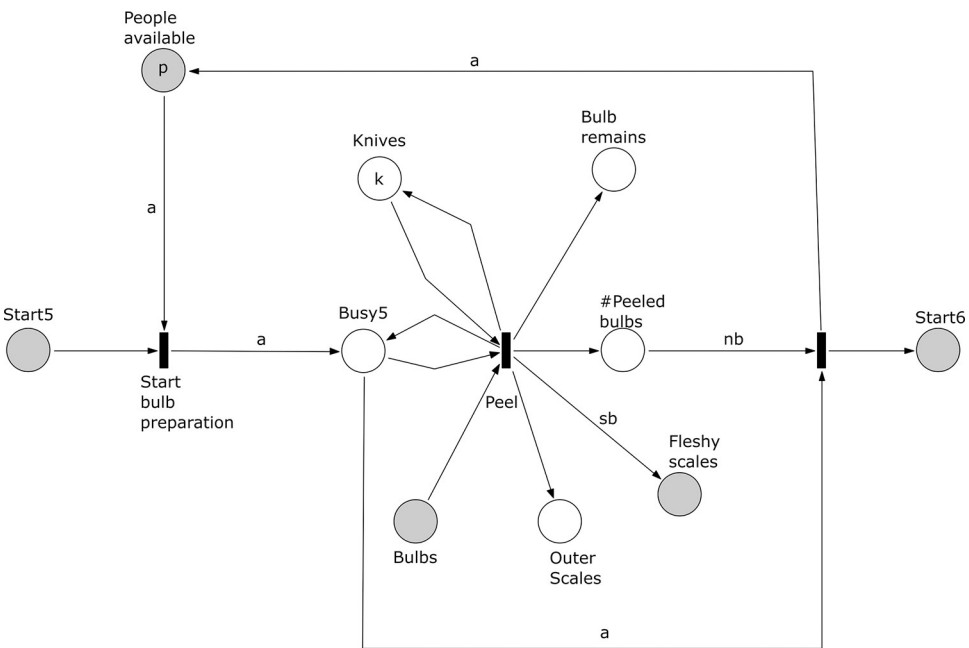

**Fig 7. Subprocess 5 Prepare Bulbs of the *A. coranica* model.**

types of blocks can occur concurrently. The subprocess ends when the number of tokens in the places '# Collected small blocks' and '# Collected large blocks' correspond with the counts of *nbs* and *nbl* blocks.

*Subprocess 5 Prepare Bulbs* (Fig 7). For this subprocess the bulbs collected in subprocess 1 are required and subprocess 1 should be completed. The subprocess 5 requires *a* people that peel the bulbs using knives. There are *k* knives available. The subprocess yields bulb remains that can be replanted; outer scales that are to be discarded; and a pile of fleshy scales that will be processed further. Each bulb produces *sb* fleshy scales. The process ends when all bulbs have been peeled.

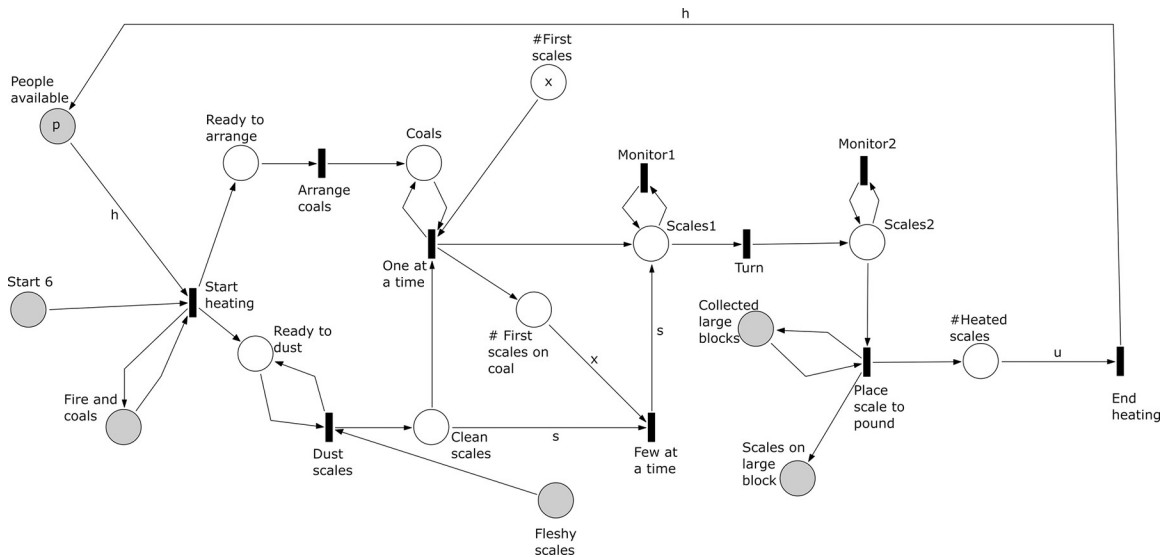

**Fig 8. Subprocess 6 Heat Selected Scales of the *A. coranica* model.**

*Subprocess 6 Heat Selected Scales* (Fig 8). This subprocess starts after the pile of fleshy scales is harvested in subprocess 5, and when the fire and coals of subprocess 3 are ready. It begins with the arrangement of the coals and dusting the piled scales. The cleaned scales are placed on top of the coals, initially *x* and later *s* at a time. In the model *s* is defined by the number of fleshy scales that each bulb produces and the number of scales that are placed first on top of the coals, which is given by the expression $s = (nb * sb) - x$. All the fleshy scales, indicated in the model as *u*, where $u = nb * sb$, are monitored while on top of the coals and turned to heat them further, and then removed and placed on one of the *nbl* calcrete blocks collected in subprocess 4. In the model each scale is placed on one of *nbl* calcrete blocks (anvils). As with the other subprocesses, multiple people (here *h*) can be involved in the subprocess. This subprocess is executed partially in parallel with subprocess 7 and 8 and finishes after all scales have been heated.

*Subprocess 7 Pound* (Fig 9). The subprocess requires scales on anvils from subprocess 6, small blocks collected in subprocess 4, and fire and coals from subprocess 3. This process starts after at least one heated scale is placed on the large calcrete block. The subprocess requires one person from the *p* people available to pound one scale. Heated scales may be cleaned before start pounding and folding. The scales are pounded using the collected small blocks as hammer stones. Note that places '# Pounded scales' and ' # Scales available for kneading' count the number of scales that went through the pounding and folding sequence. This subprocess and subprocess 6 and 8 occur partially in parallel. If activated, subprocess 7 can stop temporarily when there are no tokens in the place 'Scales on large block' and it 7 finishes when *u* scales (see subprocess 6) have gone through the pounding and folding sequence.

Subprocess 8 Knead (Fig 10). Subprocess 8 requires pounded scales from subprocess 7 and fire and coals produced in subprocess 3. This subprocess requires at least one pounded scale and at least one person available to start. The scales are first kneaded one by one until all scales are pulp. Note that the place 'All scales pounded' and the parameter *u* control that there are no scales left in subprocess 6 and 7. When all scales have been kneaded to make a pulp, they are reheated together using the fire and coals to form a pulp ball, which is kneaded again to end subprocess 8 and obtain an adhesive.

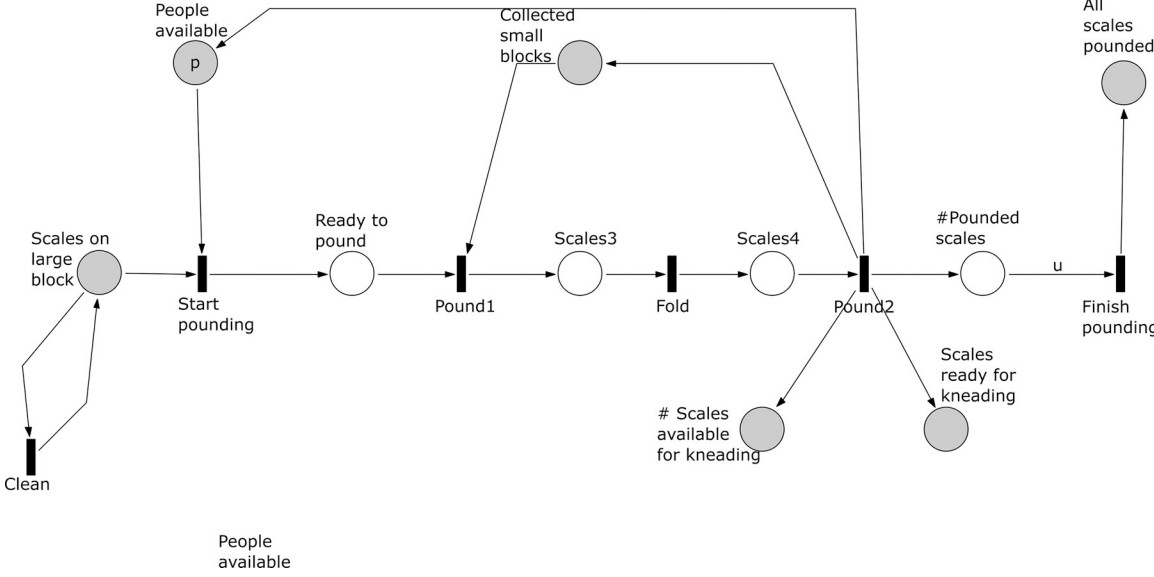

**Fig 9. Subprocess 7 Pound of the *A. coranica* model.**

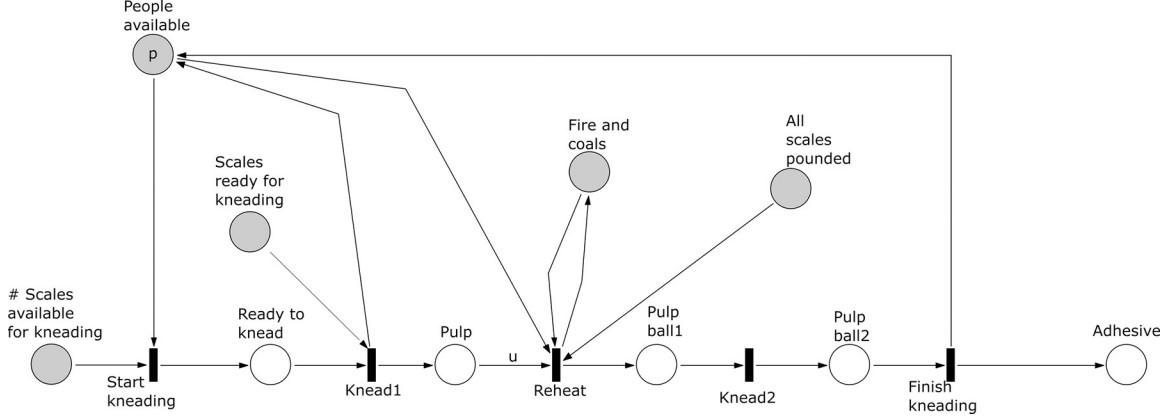

**Fig 10. Subprocess 8 Knead of the *A. coranica* model.**

The Petri net model makes use of the fact that some subprocesses can occur concurrently and order-independently to obtain the adhesive product. The complete Petri net model is included as a pnml file in the supplementary information (pnml file in S1 File). Fig 11 show the sub-processes of the *A. coranica model* represented as macro-transitions and the places connecting the subprocess highlighted in grey to capture the start and end of single sub-processes. The initial markings required to finish each subprocess and the final markings of each subprocess in the scenario with one available maker can be found in the supplementary information (Tables 1–8 in S1 Appendix).

Subprocesses 1, 2 and 4 provide resources for other subprocesses and in our model, they can occur concurrently depending on the number of people involved. Subprocess 3 depends on, and is causally preceded by subprocess 2, while subprocess 5 depends on subprocess 1. Subprocess 5 can also occur concurrently with subprocesses 2 and 4. Subprocess 2 must happen before the subprocess 3 because firewood and branches are required to light a fire. Subprocess 5 depends on the bulbs from subprocess 1 to produce fleshy scales, which are required in turn by the subprocess 6. To finish subprocesses 6 in scenario with only one maker available, subprocess 4 needs to be completed first. Subprocesses 6, 7 and 8 require inputs from other subprocesses, but they occur partly in parallel with each other.

## 3.3. A Petri net model for *O. schinzii* adhesive production

As for the *A. coranica* adhesive production, also the *O. schinzii* production system can be considered as being divided in eight subprocesses. Together, the subnets have *p* people available with a toolkit represented by *ds* digging sticks, *k* knives, *fs* firesticks, usually equal to one, and one glue carrier. The subprocesses listed below follow the order of the ethnographic description, but they are not necessarily executed in this order:

1. Dig Roots,

2. Collect Firewood,

3. Light Fire,

4. Make Applicator,

5. Root Preparation,

6. Burn and Crush Grass,

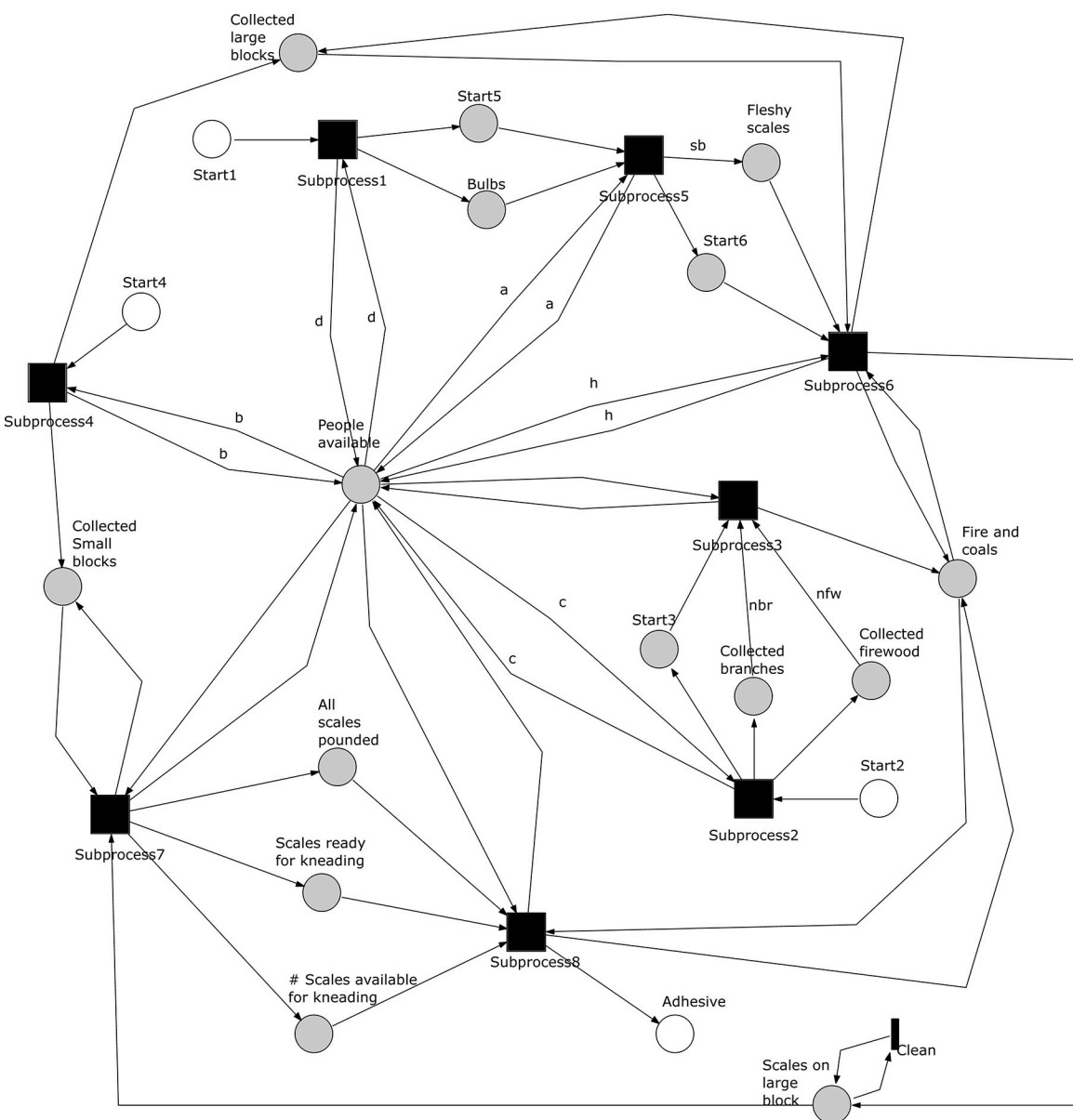

**Fig 11. Petri net showing the subprocesses of *A. coranica* model.** Subprocesses represented as macro-transitions (black squares) and the places connecting subprocesses highlighted in grey.

7. Heat Roots,

8. Dip and Mix Latex.

*Subprocess 1 Dig Roots* (Fig 12). The net for this subprocess has a place 'Start 1' initially marked with one token, to guarantee that the process is executed at most once. Roots are exposed by digging sand and *nr* roots are extracted. When all the roots have been extracted, makers use the dug sand to fill the holes around the bush. The number of required roots is marked in the place 'Roots needed' as *nr*. The place 'Hole covered' marks that the transitions 'Dig', 'Extract', and 'Fill hole' have been finished. A total of *d* people participates in this subprocess. A token in place 'Collecting roots done' marks the end of this subprocess and enables one

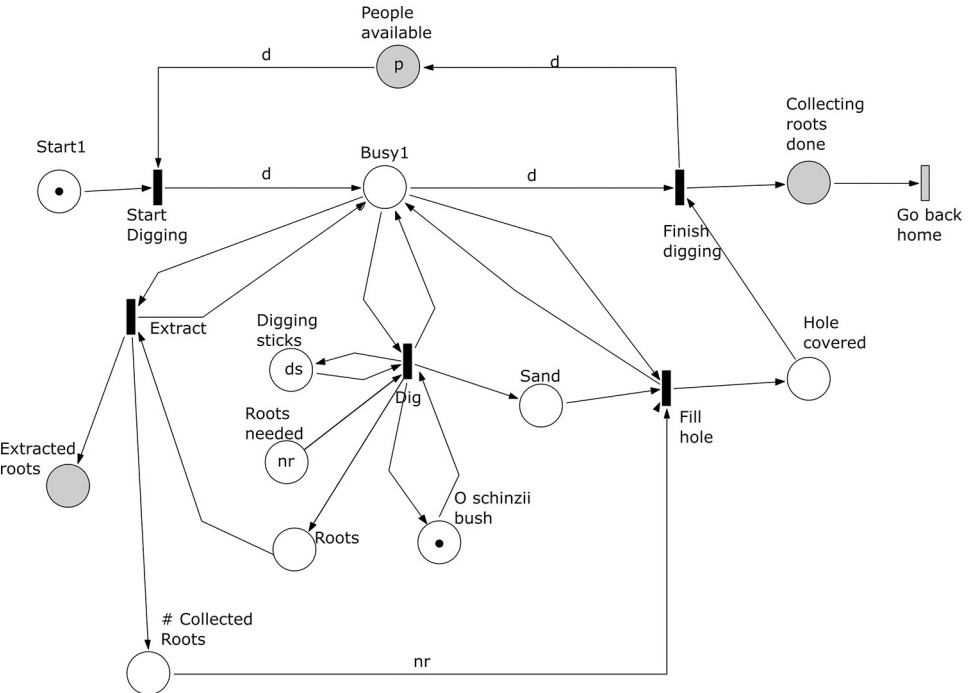

**Fig 12. Subprocess 1 Dig Roots of the *O. Schinzii* model.**

of the two conditions for changing location. In the model the change of location is represented by the transition 'Go back home'. All people involved (here *d*) go back home when they have finished subprocess 1 (and 2).

*Subprocess 2 Collect Firewood* (Fig 13). This subprocess includes a marked place 'Start2' to ensure it is executed at most once. Branches from two distinct species (*Combetrum* sp. and *T.*

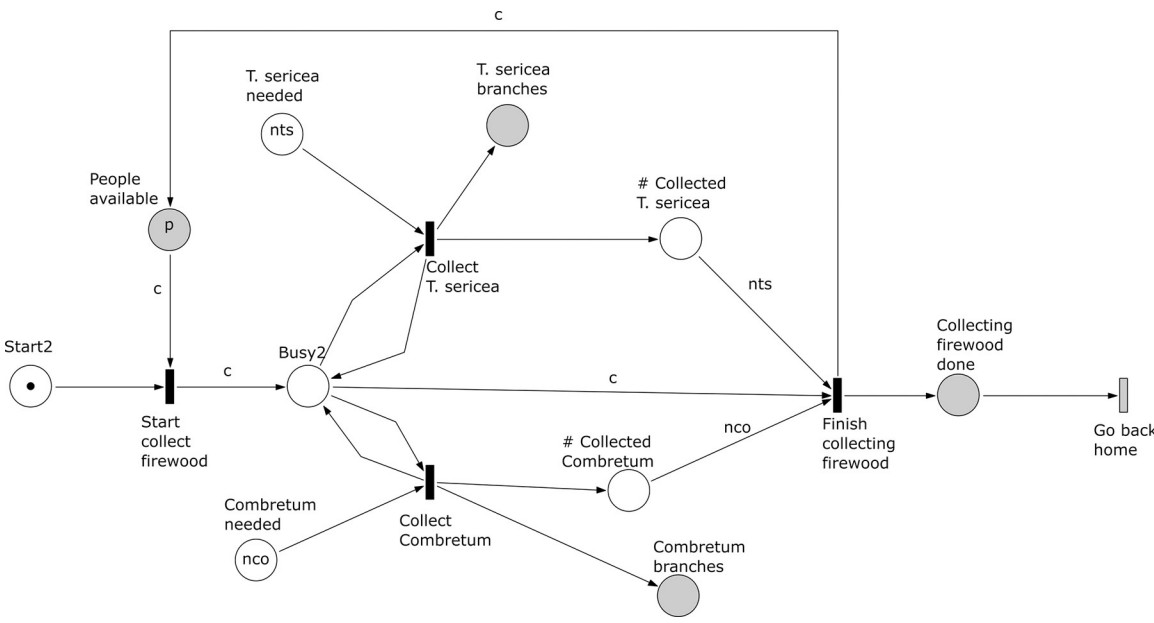

**Fig 13. Subprocess 2 Collect Firewood of the *O. Schinzii* model.**

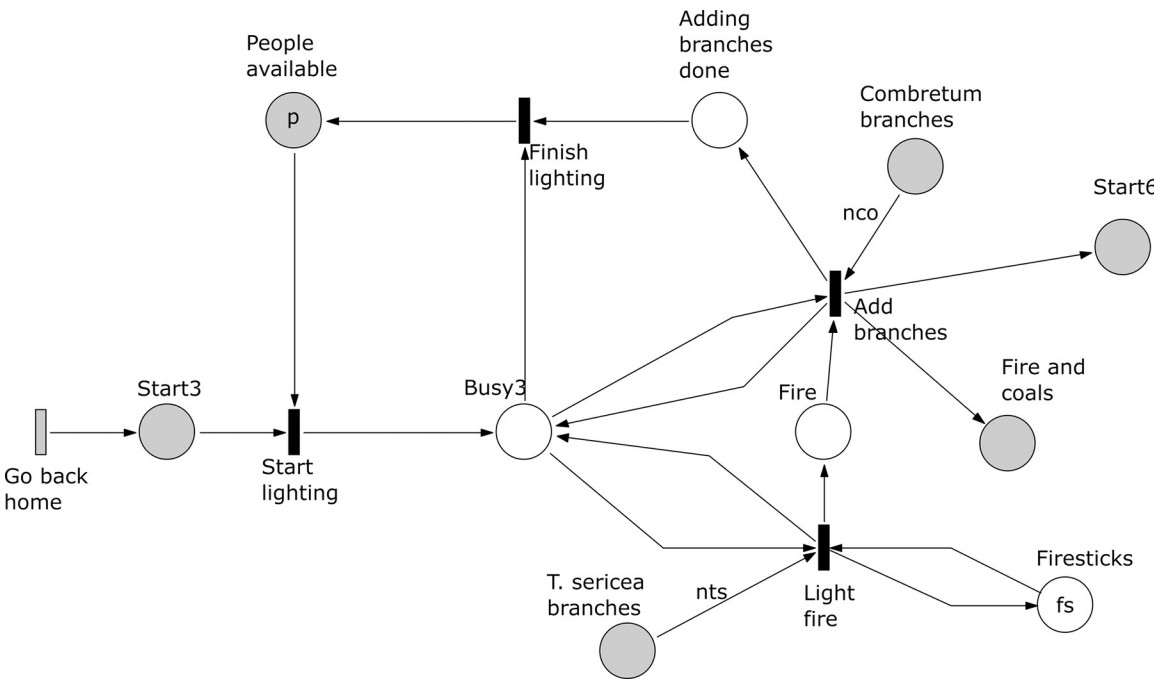

**Fig 14. Subprocess 3 Light Fire of the *O. Schinzii* model.**

*sericea*), represented as *nco* and *nts* branches, are collected to make a fire and coals. Here *c* people can collect both types of branches. The place 'Collecting firewood done' marks the end of the subprocess and it is the second condition of the transition 'Go back home'.

*Subprocess 3 Light Fire* (Fig 14). The subprocess can start after the place 'Start3' has been marked, that is after subprocesses 1 and 2 have been finished and the transition 'Go back home' has occurred. In the model the *nts* and *nco* branches collected in subprocess 2 are used here to light a fire and produce coals, respectively. The place 'Fire' controls that *nco* branches in the model are added after a fire is lit. Here, one person is involved, and requires at least one fire stick set to start the fire. Once *nco* branches are added the subprocess finishes and fire and coals are produced. The place 'Start6' controls that subprocess 6 occurs after fire and coals are available.

*Subprocess 4 Make Applicator* (Fig 15). This subprocess is enabled after the subprocesses 1 and 2 have been finished and the transition 'Go back home' have been executed. Here a *G. flava* branch is cut into an applicator by a person using a knife. The subprocess ends after producing the applicator, which will be used in subprocess 8 to transfer latex to a glue carrier.

*Subprocess 5 Root Preparation* (Fig 16). Again this subprocess starts after subprocesses 1 and 2 are finished. Here slits are cut in the roots extracted in subprocess 1. The slits are cut by *a* people using *k* knives. The roots with slits will be used by subprocess 7. The subprocess ends when all roots have been processed, which is marked in the place '# Roots with slits' with a number of tokens equal to all roots collected in subprocess 1.

*Subprocess 6 Burn and Crush Grass.* (Fig 17). Subprocess 6 starts after subprocess 3 has been completed. In this subprocess *q* people collect *ng* units of grass. The grass is lit using fire and coals and lifted to burn. The burnt grass is crushed to form a black powder. The subprocess ends after all *ng* units of grass have been crushed, which is controlled by counting the number of crushed units of grass in the place "# Crushed grass". Note that here are reachable markings at which the transitions 'Collect grass', 'Light grass', and 'Lift' are concurrently enabled.

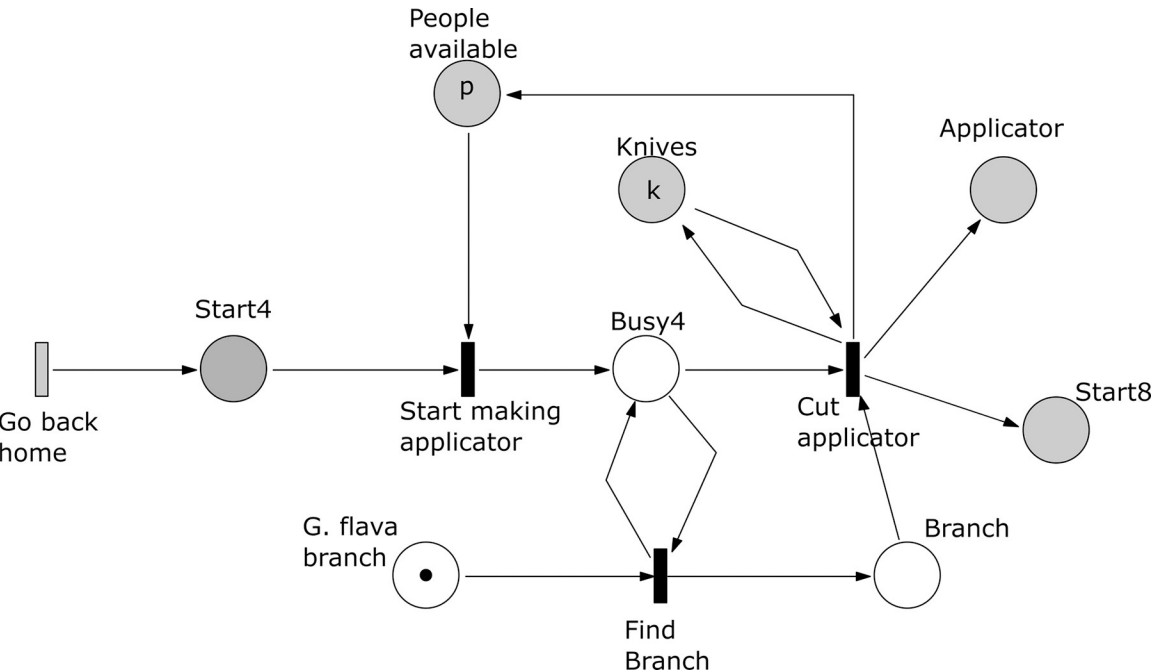

**Fig 15. Subprocess 4 Make Applicator of the *O. Schinzii* model.**

*Subprocess 7 Heat Roots* (Fig 18). This subprocess starts after subprocess 5 has been completed. The subprocess requires one person, the roots with slits produced in subprocess 5, and the fire and coals from subprocess 3. Coals are arranged at the edge of the fire and the roots are laid one at a time on these coals. As a result, latex exudates from the roots. The subprocess has a place '# Heated roots' that controls that all roots required are heated before finishing this subprocess. The subprocess finishes when all *nr* roots have produced latex.

*Subprocess 8 Dip and Mix Latex* (Fig 19). This subprocess can occur partially in parallel with subprocesses 6 and 7 if the number of people available is at least *q* + 2 and subprocess 4

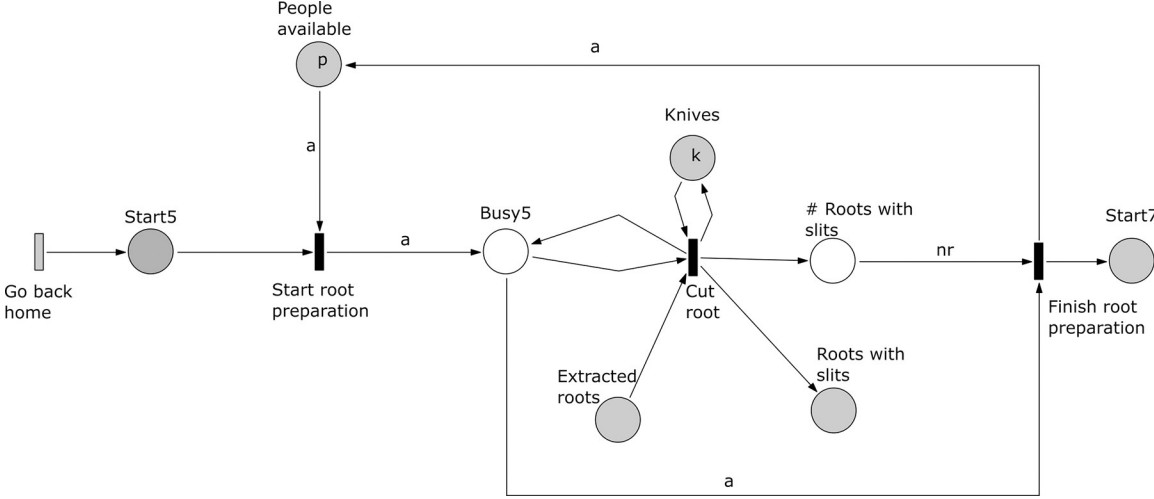

**Fig 16. Subprocess 5 Root Preparation of the *O. Schinzii* model.**

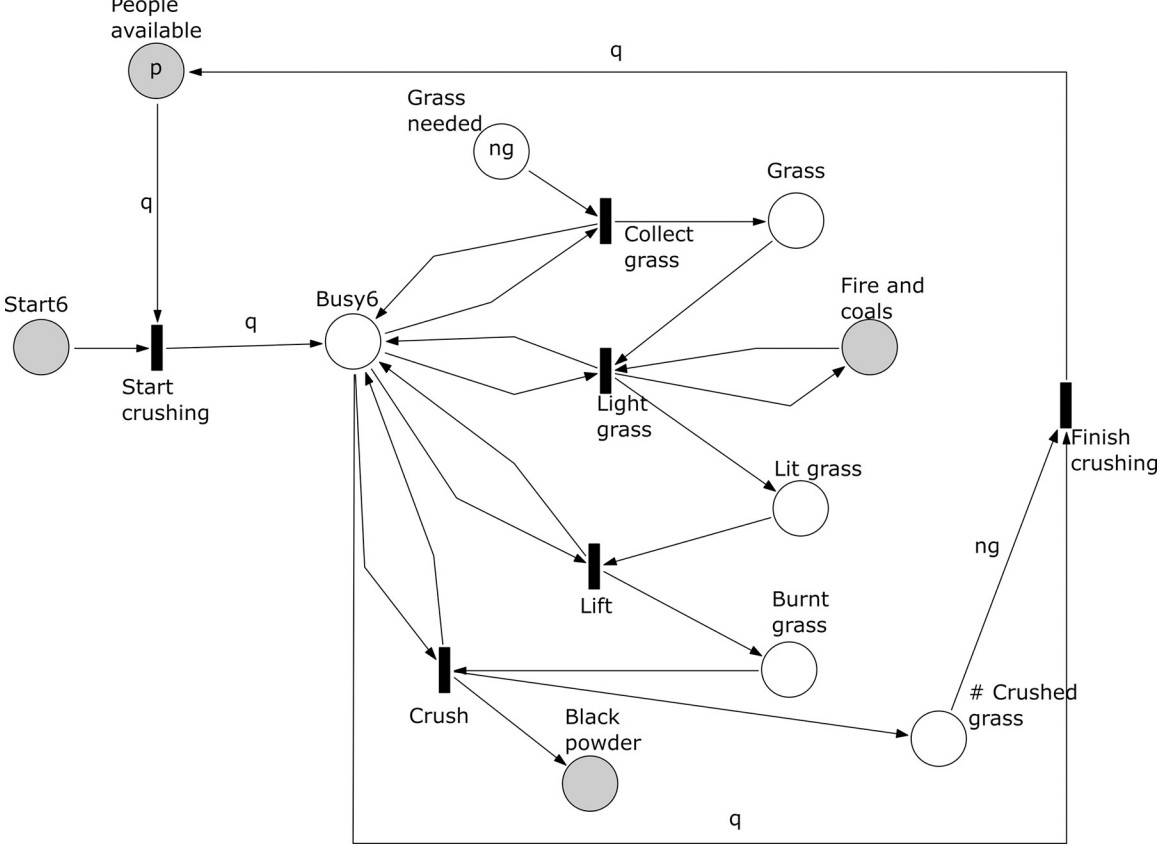

**Fig 17. Subprocess 6 Burn and Crush Grass of the *O. Schinzii* model.**

has been finished. Subprocess 8 requires *m* people, a glue carrier from the toolkit, the applicator produced in subprocess 4, roots with latex from subprocess 7 and black powder produced in subprocess 6. In the model there will be eventually *nr* roots with latex. Latex is extracted

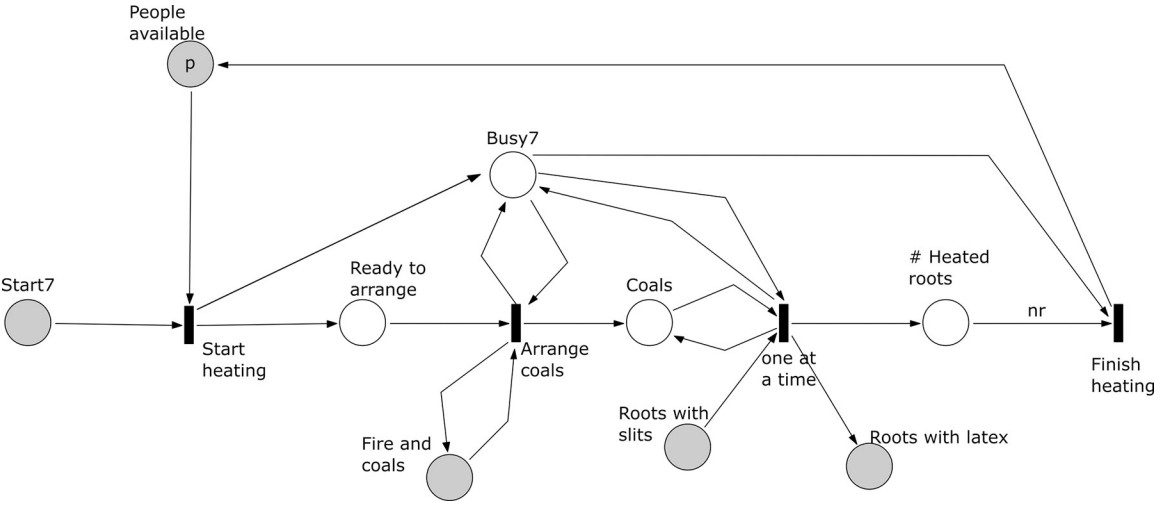

**Fig 18. Subprocess 7 Heat Roots of the *O. Schinzii* model.**

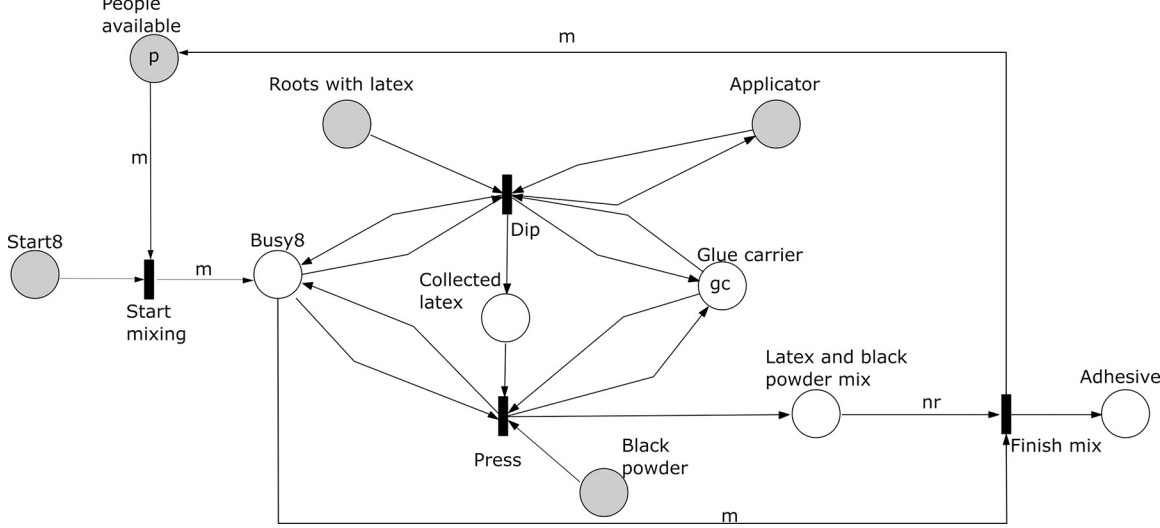

**Fig 19. Subprocess 8 Dip and Mix of the *O. Schinzii* model.**

from the roots using the applicator and transferred to the glue carrier. The black powder is pressed in layers on the glue on the carrier to make a black adhesive. Note that in the model the makers can collect first all the latex and then press all the black powder or alternate between both actions and layers of latex and black powder. The process is completed when all the latex has been collected into the carrier and all black powder has been pressed into the latex. The final output is the adhesive which also ends the production process.

The *O. schinzii* model shows relations that allow concurrency in the execution of the sub-processes. The complete Petri net model is included as a pnml file in the supplementary information (pnml file in S2 File). Fig 20 show the sub-processes of the *O. schinzii model* represented as macro-transitions and the places connecting the subprocess highlighted in grey to capture the start and end of single sub-processes. The initial markings required to finish each subprocess and the final markings of each subprocess in the scenario with one available maker can be found in the supplementary information (Tables 9–16 in S1 Appendix). The potential for concurrency is controlled by a change in location after the procurement of two of the material resources. The subprocesses 1 and 2 are executed at the first location and they can be executed in any order. The other six subprocesses are executed at a different location. Without a change in location and enough makers available, the subprocess 4 may occur concurrently with subprocesses 1 and 2. The subprocess 3 can also be executed concurrently with other subprocesses when enough people are involved in the production system. The other sub-processes in the second location require inputs from subprocesses 1 and 2.

## 3.4. Simulations

To gauge behavioural complexity of the production systems, we computed the reachability graphs for both models with different numbers of makers available for executing the processes. We also calculated the reachability graphs for a variant of the *O. schinzii model* without the transition 'Go back home'. To make this variant, we deleted the transition 'Go back home' and the places 'Start5' and 'Start3'. Then, we connected to connect the place 'Collecting roots done' (subprocess 1) to the transition 'Start root preparation' (subprocess 5) and the place 'Collecting firewood done' from subprocess 2 to the transition 'Start lighting' (subprocess 3). We also added a token to the initial marking of the place 'Start4' to ensure that subprocess 4 occurs at

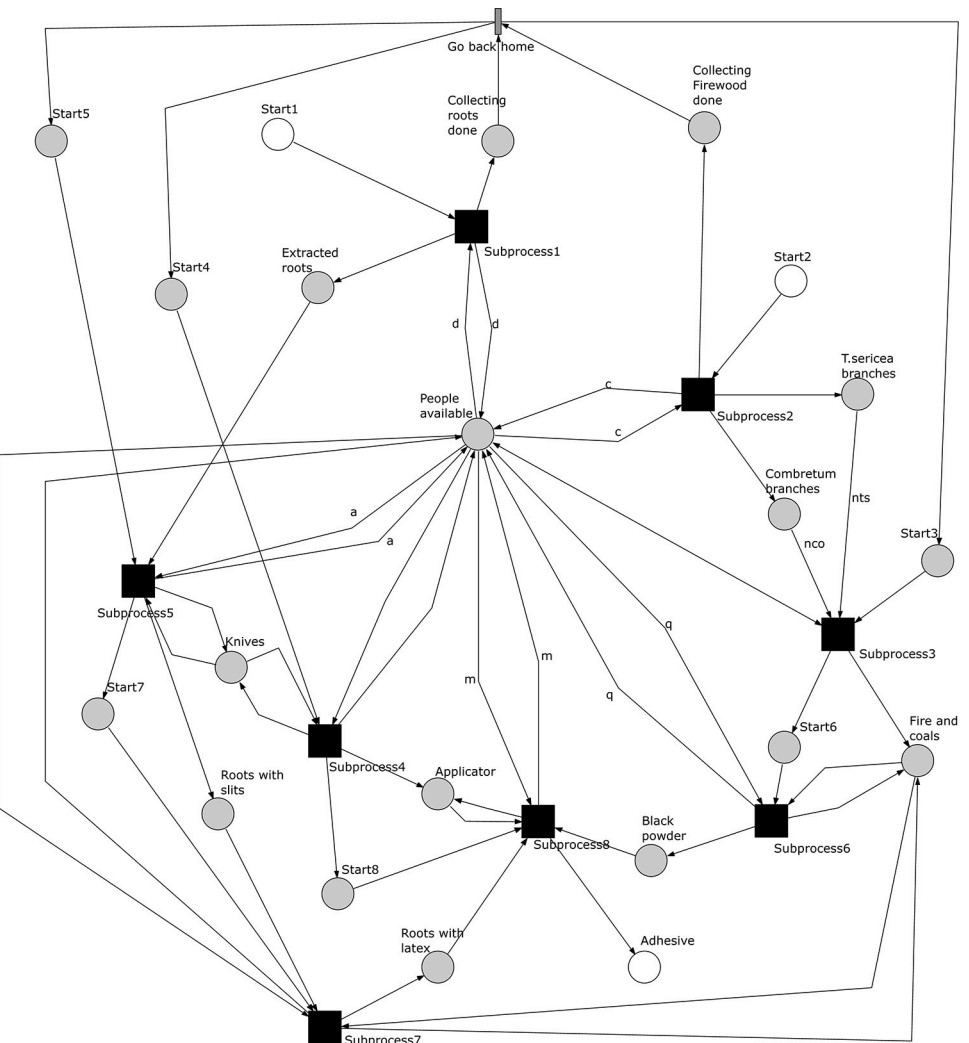

**Fig 20. Petri net showing the subprocesses of *A. coranica* model.** Subprocesses represented as macro-transitions (black squares) and the places connecting subprocesses and the transition go back home are highlighted in grey.

most once. This variant of the Petri net model for the *O. schinzii* production system is included as a pnml file in the supplementary information (pnml file in S3 File).

We compared the state space size of the reachability graphs of the Petri net models as proxy for the behavioural complexity of the production processes. We also compared the minimum number of available makers in the simulations that generated the maximum state space sizes of the processes with the ethnographic descriptions about the size of the Ju/'hoan hunting and collecting parties. We increased the number of makers by one until the state space of the models did not increase further. The values used for all variables are presented in the supplementary information (Table 17 in S1 Appendix).

We produced pnml files with Snoopy software [64] for each initial marking of the Petri net models. These pnml files (see supporting information) were used to calculate the reachability graphs with the TINA toolbox for Petri nets [65]. We computed a total of thirty-three reachability graphs: eleven for the *A. coranica* model, eleven for the complete *O. schinzii* model; and eleven the *O. schinzii model* without the transition 'Go back home'.

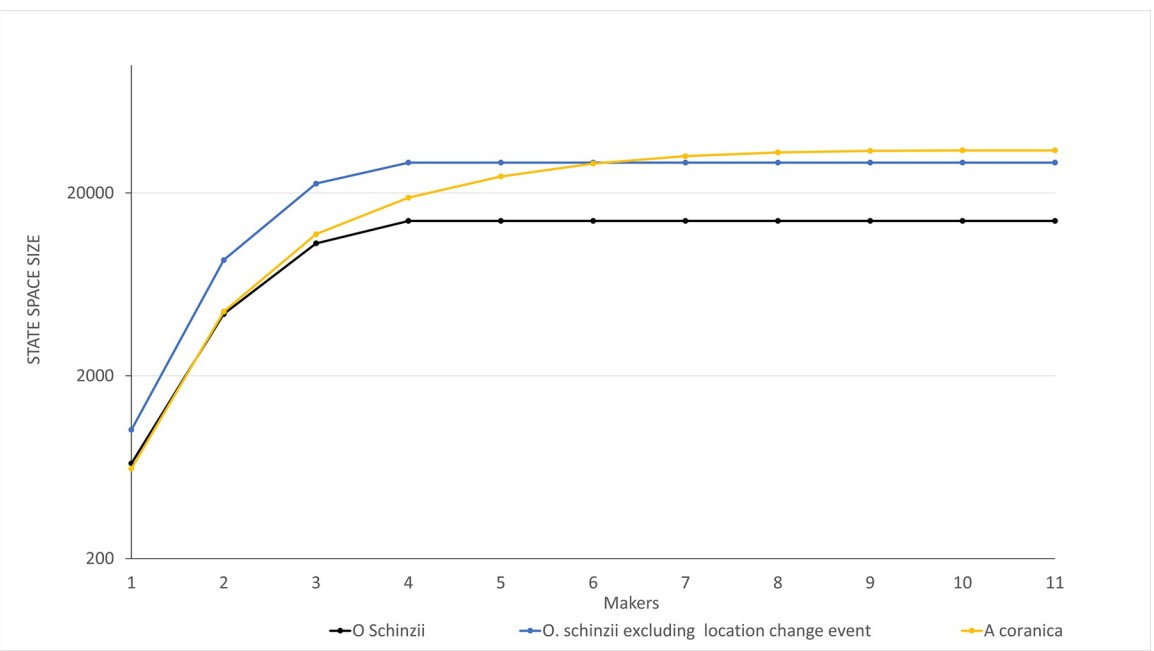

**Fig 21. Comparison of the state space of the models.** The lines represent the states spaces for the simulations of the *A. coranica* (yellow) and *O. schinzii* (black) models and the effect of deleting the transition 'Back home' associated with the location change from the *O. schinzii* model (blue). The state space size (vertical axis) is plotted on a log scale. The number of makers of each simulation (horizontal axis) is plotted on a linear scale.

In the scenario with one maker (*p = 1*), the *A. coranica* model had the lowest number of reachable states with 621 and the *O. schinzii* model showed 663 states. The *O. schinzii* model without transition 'Go back home' had 1203 reachable states. The results of the simulations showed a strong increment in the number of reachable states after a second maker was available (p = 2), and the number of reachable states for the *A. coranica* model *(N = 4488)* and the *O. schinzii* model *(N = 4350)* were similar (Fig 21). Compared with these two models, the variant of the *O. schinzii* model without a location change almost doubled the number of reachable states when two makers were available (N = 8597). The number of reachable states for the complete *O. schinzii* model (N = 14074) and the variant without the transition 'Go back home' (N = 29333) did not increase anymore once four makers (*p ≥ 4*) were available. The increase in the number of reachable states of the *A. coranica* model continued until eleven were made available (N = 34228).

## 4. Discussion

### 4.1. Petri nets to model ancient and traditional socio-technical systems

Our study aimed to introduce Petri nets and to resolve analytical challenges in the study of production systems. First, we sought to keep away from describing production systems as sequences of events. For example, [25, 58, 67] rely on natural language to describe production processes. Such descriptions require sequencers such as 'then', 'next', and 'after', to structure the observed events. Instead we focus on the relations between events and resources using the Petri net formalism and we used them as equivalent components to structure the models. This allowed us to characterize the structure of the production process and simulate the behaviour of the system. The formal definitions and the intuitive graphical notation of Petri nets tackled the problems generated by unsystematic representations of current approaches. We also

showed how basic concepts such as reachability graphs can be used to measure and compare complexity between production systems and the effects of changes in resources and events.

## 4.2. Behavioural complexity and concurrency of the models

Several actions in the production of *A. coranica* and *O. schinzii* adhesives can occur concurrently. The identification of potential for concurrency in both models indicate that Petri nets are a promising framework to model and analyse traditional and ancient production systems. These findings suggest that non-industrialized societies produced adhesive technologies with order-independent and concurrent behaviours that may be prevalent in their production systems. This hypothesis is supported by other concurrent behaviours identified in the production of other traditional compound adhesives [58].

The complexity generated by potential concurrent activities in the two adhesive production systems modelled seems to be due to the processing of resources rather than the collection of diverse types of resources. Compound adhesives such as the *O. schinzii* adhesive are described as more complex than single component adhesives like the *A. coranica* adhesive. However, the state space sizes of the reachability graphs suggest that the behavioural complexity associated with the *A. coranica* adhesive production system is similar to the behavioural complexity of the *O. schinzii* adhesive production. Moreover, when more individuals are available to participate in the process, the state space of the *A. coranica* model tends to be larger than the *O. schinzii* model (Fig 21). These findings suggest that the number of components in a technology cannot be used as unique indication of complexity. The behavioural complexity of a production system of single component technology may also be high due to the nature of events during processing.

The results suggest that some of the complexity in the procurement of the *O. schinzii* system is related to a location change event. This corresponds with the collection of *O. schinzii* roots, which occurs in a place near, but different, from the home of the makers. The ethnographic account describes the collection of the grass (*A. adscensionis*) to make the black powder as occurring in the residence of the makers. *A. adscensionis* is widely distributed in the southern Africa arid and semi-arid environments [68], which indicates *A. adscensionis* specimens would be likely found in the same locations as *O. schinzii* roots. The change in location, therefore, is not well explained by differences in geographical location of the components. The ethnographic account does not give details on why this location change occurred, but tentatively we can suggest that logical moves may have helped to lower the behavioural complexity of the production process. Makers could prefer to distribute activities of the production process between two or more places, in the same way that mobile groups solve several problems by moving across their territories [69]. Changes in location during a production process divide events in stages, reducing the information that needs to be processed at a given time. The *O. schinzii* adhesive is used in a larger production process to obtain poisoned arrows, which can be produced with a great diversity of components, and it is a potentially dangerous activity that may require several cognitive abilities [70]. This suggests that the location change in the production of *O. schinzii* adhesive may be a strategy to lower the behavioural complexity of the adhesive production and its use in other production processes like the production of poisoned arrows. Lowering the behavioural complexity of producing the adhesive material may help to reduce the risk of making possibly fatal mistakes during the use of adhesive technology in other production processes. Previous studies have shown that the risk of failure of resources is an important driver of the diversity and complexity of technological behaviours [71, 72]. Here we suggest that risk of failure of producing a technology or the risk in production processes with multiple and potential harmful components may also steer technological behaviours and

trigger the inclusion of control activities in the production process to reduce the behavioural complexity of the processes.

## 4.3. Maximization concurrency potential

The results suggest that executing the production systems with arbitrarily large groups of makers might be a costly strategy. Adding new individuals, in general lead to an increase of the state space size of the reachability graphs. The largest increment in the number of states occurred after a second individual was made available to conduct the activities. The increase in number of reachable states stopped after four and eleven individuals were added to the *O. schinzii* and *A. coranica* models, respectively. However the increase in the number of reachable states of the *A. coranica* model after introducing the fifth maker occurred at low rate, showing that the number of possible concurrent activities was only somewhat different than with the scenario with lower number of makers. It can thus be suggested based on the Petri net models, that for real production scenarios with two to four individuals available facilitate gaining the benefits in flexibility and parallel execution derived from the concurrency potential. These scenarios may also reduce the disadvantages generated by communication and synchronization of activities and use of tools generated when more people involved in the real processes. This is consistent with the ethnographic of the Ju'/hoansi where hunting and other activities in which tools with adhesives are used tend to be executed by individuals, pairs, and sometimes small groups [73].

## 4.4. Further research

Further research can explore at least Four topics. The first possibility is to use Petri nets to reconstruct the production processes of technologies found in the archaeological record. For this, detailed ethnographic analogies, controlled experiments, and reconstructions of the materials and past use of the objects will be required. The second possibility is to study how much of the complexity documented in the state spaces of the production systems is perceived by the makers and how production systems are affected by deterministic and stochastic decision-making processes. New ethnographic data could be used to answer these questions. Approaches like quantitative ethnography [74] that generate substantial amounts of structured data are a good choice for such studies. A Third path for further research is to explore how to measure specific aspects of human behaviour such as cultural transmission or cognitive load using the reconstructions of past production process as Petri net models. Analytical methods of Petri nets such as invariants, inequalities, and distributed runs might be used as proxies to link those aspects with the structure or behaviour of production systems. Finally, studying the effects of variations of available makers in the internal dynamics of the subprocesses is other future direction of research. We explored in the models and simulations the effect of the variation of p in the dynamics between subprocesses, but the internal dynamics of many sub-processes are independent of the variation p. One can use the local variables assigned to the makers involved in a subprocess to study the effects of the internal dynamics of the subprocess in the overall complexity of the production system.

## 5. Conclusion

Production systems encapsulate how humans create technology through interactions between resources and events. The structure and behaviour of such systems demand suitable models to ensure a comprehensive representation, a crucial step before assessing the complexity of the system becomes possible. In this paper we demonstrated for two particular technological processes how Petri nets are a plausible solution for several challenges faced when analysing

complexity with sequential models. We also showed how reachability graphs can be used to measure behavioural complexity and explore the effects of changes in the system's structure on system behaviour. The results suggest that the complexity of ancient technologies can be attributed to multiple factors. Measuring complexity in terms of 'more' or 'less' is inadequate to understand the implications of those factors for human behaviour. Rather a comprehensive set of measurements of complexity for ancient technologies should link measurements with how humans may solve the problem of producing a given technology. Considering current debates about the complexity of technological systems and the need of understanding the differences between production processes of ancient technologies and their implications for past societies, Petri nets are a valuable addition to the set of methodologies for studying dynamics of ancient production systems.

## Supporting information

**S1 File. Pnml file representing the *A. coranica* model.**
(PNML)

**S2 File. Pnml file representing the *O. schinzii* model.**
(PNML)

**S3 File. Pnml file representing the *O. schinzii* model without the transition 'Go back home'.**
(PNML)

**S4 File. csv file including data of the reachability graphs.**
(CSV)

**S1 Appendix. Tables.** Section A) initial markings and final markings of the subprocesses of the *A. coranica* model (Tables 1–8) and the *O. schinzii* model (Tables 9–16). Section B) Variables for the *A. coranica* and *O. Schinzii* models used to calculate the reachability graphs (Table 17).
(DOCX)

## Acknowledgments

We thank Pierre Mercuriali, Sylvia Mota De Oliveira, Paul Kozowyk and Alessandro Aleo for their comments, suggestions, and insightful discussions.

## Author Contributions

**Conceptualization:** Sebastian Fajardo, Jetty Kleijn, Frank W. Takes, Geeske H. J. Langejans.

**Data curation:** Sebastian Fajardo, Jetty Kleijn, Frank W. Takes, Geeske H. J. Langejans.

**Formal analysis:** Sebastian Fajardo, Jetty Kleijn, Frank W. Takes, Geeske H. J. Langejans.

**Funding acquisition:** Geeske H. J. Langejans.

**Investigation:** Sebastian Fajardo, Jetty Kleijn, Frank W. Takes, Geeske H. J. Langejans.

**Methodology:** Sebastian Fajardo, Jetty Kleijn, Frank W. Takes, Geeske H. J. Langejans.

**Supervision:** Jetty Kleijn, Frank W. Takes, Geeske H. J. Langejans.

**Validation:** Sebastian Fajardo, Jetty Kleijn, Frank W. Takes, Geeske H. J. Langejans.

**Visualization:** Sebastian Fajardo, Jetty Kleijn, Frank W. Takes, Geeske H. J. Langejans.

**Writing – original draft:** Sebastian Fajardo, Jetty Kleijn, Frank W. Takes, Geeske H. J. Langejans.

**Writing – review & editing:** Sebastian Fajardo, Jetty Kleijn, Frank W. Takes, Geeske H. J. Langejans.

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
