## [Decision Letter · Decision Letter 0]

21 Sep 2022

PONE-D-22-18669Modelling and measuring complexity of traditional and ancient technologies using Petri netsPLOS ONE

Dear Dr. Fajardo,

Thank you for submitting your manuscript to PLOS ONE. After careful consideration, we feel that it has merit but does not fully meet PLOS ONE’s publication criteria as it currently stands. Therefore, we invite you to submit a revised version of the manuscript that addresses the points raised during the review process.

We look forward to receiving your revised manuscript.

Kind regards,

Fabrizio Pecoraro

Academic Editor

PLOS ONE

Journal Requirements:

Reviewers' comments:

Reviewer's Responses to Questions

**Comments to the Author**

1. Is the manuscript technically sound, and do the data support the conclusions?

Reviewer #1: Yes

Reviewer #2: Yes

2. Has the statistical analysis been performed appropriately and rigorously? 

Reviewer #1: N/A

Reviewer #2: N/A

3. Have the authors made all data underlying the findings in their manuscript fully available?

Reviewer #1: Yes

Reviewer #2: Yes

4. Is the manuscript presented in an intelligible fashion and written in standard English?

Reviewer #1: Yes

Reviewer #2: Yes

5. Review Comments to the Author

Reviewer #1: The authors propose a net modeling approach (Petri nets) to the measurement of artifact production complexity and apply their measure to the production of adhesives (specifically, two types of adhesives) by a hunter-gatherer group in southern Africa. I think the paper represents a substantive contribution to the literature on artifact complexity and its measurement and I have no major criticisms of the manuscript.

My own approach to the measurement of artifact complexity has been grounded in computation theory with reference to classic models of computation (e.g., finite state machine/grammar), but I think the use of Petri nets has some advantages as a high-resolution analytical tool.

The production of adhesives has a deep history in the Paleolithic, so the measurement of adhesive production complexity has interesting implications for human cognitive faculties in a relatively early time range (>300 ka). Diagrammed out with Petri nets, the production process, which likely was similar during Paleolithic times, is significantly more complex, in terms of the number and variety of steps, than most archaeologists would imagine. The authors’ approach effectively illustrates this point with their net diagrams.

One of the only criticisms that I have of the current paper is that I think the critique of the chaîne opératoire approach is overdone (and may unnecessarily alienate some readers). I suggest that the authors adopt a different tone and approach, arguing for the advantages of using Petri nets, rather than hammering on the deficiencies of the chaîne opératoire, which I think they are overstating

Finally, I note that one of the most important implications of the significantly increased computational complexity of artifact design and production during the Paleolithic is increased memory storage capacity. The more powerful computation models reflect this phenomenon: a Turing machine is simply a finite-state machine with an infinite memory tape. I think this point is well illustrated by the complex process of adhesive production, which implies considerable memory storage.

Reviewer #2: Firstly, I would like to point out that my research topics are not focused on traditional and ancient technologies. However, for this article I have studied the main topics covered by the authors such as the Ammocharis coranica process, the chaine opératoire.

On the other side, my activities perfectly match with the modelling part of the paper where modelling methodologies are applied to describe business and production processes. For this reason this review is mainly focused on the syntactical aspect of the paper and in particular in the use of Petri nets.

Hereafter are reported my main suggestions:

Considering the methodology (paragraph 5):

1) It is not explained what the symbol E represents in the formal definitions reported

2) These formal definitions are purely academic and are not necessary for the purpose of the paper as they are not used to describe properties of the representation of the production model that is based on a classical Petri net without particular constraints (basic axioms).

3) Considering that the paper may have different types of potential readers (also not technicians) I suggest to present the PN in natural language. It could be more useful especially for the concepts of reachable state space.

4) In line 239 authors should report (and confirm) that: a) the firing of a transition is instantaneous; b) the choice of which transitions to shoot is random. These information are relevant to better describe the competitiveness property of the network.

Considering paragraph 3: Two Ju / ’hoan adhesive productions

1) Marking: In the analysis of the various sub-processes, a fundamental role is played by the markings, especially considering the places highlighted in grey. They represent the connection between the different sub-processes and reporting in each sub-process the initial marking for the sub-process is important. Clearly, this comment is relevant also considering the final marking of a sub-process. Therefore, I suggest to report for each (grey) place the marking which, for instance, can be represented in square brackets. This highlights the final marking of a sub-process and consequently the initial marking of the related/subsequent sub-process.

2) Overall representation: Authors should report an overall representation of the entire production process. This perspective is strictly important to facilitate readers on how the various sub-processes are connected to each other’s (e.g. precedence, competition) and therefore to understand the overall process. To accomplish this task I suggest two possible solutions: 1) A Pert model where each node is a sub-process and the edges indicate the precedence between them; 2) A Petri net where sub-processes represented as macro-transitions and where grey places are highlighted to capture the start and end a single sub-process. In this way the connections between sub-processes (the places in grey) with the relative markings are clearly highlighted; 3) An overall Petri net reported as a supporting information file, added to the main manuscript and uploaded in the PlosOne website.

3) Individual sub-processes: a) the presence of redundant situations should be better motivated. For instance, in Figure 8 the transition scales1 -> turn-> scales2 may be reduced to scales1 with the elimination of the turn transition; b) once the sub-process 7 (see Figure 9) is activated it never stops; c) the initial and final markings of each sub-process is not always straightforward. This is mainly due to the absence of the marking in the relevant model. It makes it difficult to analyse the dynamics of the models.

4) The simulation is based on the number of people available (p = 1 to 11) but with the value at 1 of the markers involved in the sub-process combined with the lack of an overview of the whole process, it is difficult to understand where the variations of p affect the space of reachable states; in fact the dynamics of many sub-processes is independent of the variation of p.

6. PLOS authors have the option to publish the peer review history of their article (what does this mean?). If published, this will include your full peer review and any attached files.

Reviewer #1: **Yes: **John F Hoffecker

Reviewer #2: No

---

## [Author Response · Author response to Decision Letter 0]

5 Oct 2022

Academic editor request #1

Journal requirement: Please ensure that your manuscript meets PLOS ONE's style requirements, including those for file naming. 

Authors’ response: We have followed the PLOS ONE’S style and format in the revised manuscript.

Academic editor request #2

Journal requirement: Please include a complete copy of PLOS’ questionnaire on inclusivity in global research in your revised manuscript. Our policy for research in this area aims to improve transparency in the reporting of research performed outside of researchers’ own country or community. The policy applies to researchers who have travelled to a different country to conduct research, research with Indigenous populations or their lands, and research on cultural artefacts. The questionnaire can also be requested at the journal’s discretion for any other submissions, even if these conditions are not met. Please find more information on the policy and a link to download a blank copy of the questionnaire here: https://journals.plos.org/plosone/s/best‐practices‐in‐research‐reporting. Please upload a completed version of your questionnaire as Supporting Information when you resubmit your manuscript.

Authors’ response: We answered PLOS’ questionnaire on inclusivity in global research (see attached form). The analysis was done based on ethnographic descriptions published by Wadley et al 2015 (DOI:/10.1371/journal.pone.0140269.g00). No personal or sensitive data was used in this study. Wadley et al. reported in the section “Background to the Ju/’hoan San of Nyae Nyae” the informed consent and ethics review process for their own research.

Reviewer #1 concern #1

Reviewer’s comment: One of the only criticisms that I have of the current paper is that I think the critique of the chaîne opératoire approach is overdone (and may unnecessarily alienate some readers). I suggest that the authors adopt a different tone and approach, arguing for the advantages of using Petri nets, rather than hammering on the deficiencies of the chaîne opératoire, which I think they are overstating

Author’s response: We agree with the suggestion of reviewer #1 that a focus on the advantages of Petri nets rather than the challenges of current available methods would make the paper more appealing to some readers. We deleted the former section “1.1 current approaches to analyse the complexity of production systems “ and rewrote former section “1.2 Petri nets: non-sequential formal models” (now 1.1) to focus on the advantage of Petri nets. Additional changes for consistency were implemented in the abstract, discussion and conclusion sections. We consider that these changes improve the readability and focus of the paper.

Reviewer #2 concern #1

Reviewer’s comment: Considering the methodology (paragraph 5): 1) It is not explained what the symbol E represents in the formal definitions reported.

Authors’ response: We now explain what symbol E represents, thanks to the correct observation of reviewer#2, in this way (marked in red): 

“The reachability graph of (N, M) is the initialised, arc labelled, directed graph 

RG(N, M) = (R(N, M),E,M )) with set of nodes R(N, M), initial node M, and set of edges E ⊆ {(M1 , t, M2): M1 , M2 in R(N, M) and M_1 □(→┴t ) M_2}.”

Reviewer #2 concern #2

Reviewer’s comment: Considering the methodology (paragraph 5): 2)These formal definitions are purely academic and are not necessary for the purpose of the paper as they are not used to describe properties of the representation of the production model that is based on a classical Petri net without particular constraints (basic axioms). 3) Considering that the paper may have different types of potential readers (also not technicians) I suggest to present the PN in natural language. It could be more useful especially for the concepts of reachable state space.

Authors’ response: We agree with the reviewer that natural language can facilitate the reading of the formal definitions and the paragraphs in section 2.1. were written with this aim. Nevertheless, we consider that the formal definitions provide clarity about the modelling approach and avoid ambiguities. Moreover, they are required to introduce the topic to a complete new set of potential readers in social sciences and humanities. We used the basic axioms with the aim of establishing the paper as reference material not only for our further work with Petri nets, but also for other potential researchers interested in implementing this modelling approach. Following the suggestion of the reviewer we incorporated an additional text to describe better in natural language the concept of reachable state space, like this (marked in red):

In the reachability graph of a Petri net, each node is a reachable marking and thus represents a possible state of the modelled system. All these possible states together form the state space of the system. 

Reviewer #2 concern #3

Reviewer’s comment: Considering the methodology (paragraph 5): 4) In line 239 authors should report (and confirm) that: a) the firing of a transition is instantaneous; b) the choice of which transitions to shoot is random. These information are relevant to better describe the competitiveness property of the network.

Authors’ response: We incorporated in the revised version the observation of the reviewer, as below (marked in red):

Consequently, the occurrence of a transition is by local enabling and has a local effect. Often the occurrence of transitions is referred to as 'firing'. Points 2 and 3 in Definition 3 together constitute the firing rule of Petri nets. The firing of a transition is instantaneous and the choice of which transition to fire when several transitions are enabled at the same time is random. When several transitions are enabled by a marking with sufficient tokens to fulfil the input requirements of each transition, then these transitions may occur concurrently at that marking. 

Reviewer #2 concern #4

Reviewer’s comment: Considering paragraph 3: Two Ju/’hoan adhesive productions: 1) Marking: In the analysis of the various sub‐processes, a fundamental role is played by the markings, especially considering the places highlighted in grey. They represent the connection between the different sub‐processes and reporting in each sub‐process the initial marking for the sub‐process is important. Clearly, this comment is relevant also considering the final marking of a sub‐process. Therefore, I suggest to report for each (grey) place the marking which, for instance, can be represented in square brackets. This highlights the final marking of a sub‐process and consequently the initial marking of the related/subsequent sub‐process.

Author’s response: We included in the supporting information an appendix with tables with the initial markings and the final markings of each subprocess in the scenario with one available maker (p =1). 

Reviewer #2 concern #5

Reviewer’s comment: Considering paragraph 3: Two Ju/’hoan adhesive productions: 2) Overall representation: Authors should report an overall representation of the entire production process. This perspective is strictly important to facilitate readers on how the various sub‐processes are connected to each other’s (e.g. precedence, competition) and therefore to understand the overall process. To accomplish this task I suggest two possible solutions: 1) A Pert model where each node is a sub‐process and the edges indicate the precedence between them; 2) A Petri net where sub‐processes represented as macro‐transitions and where grey places are highlighted to capture the start and end a single sub‐process. In this way the connections between subprocesses (the places in grey) with the relative markings are clearly highlighted; 3) An overall Petri net reported as a supporting information file, added to the main manuscript and uploaded in the PlosOne website.

Author’s response: We agree with reviewer #2 that the representations suggested can enhance the understanding of the overall process. We followed the suggestion #2 and included two Petri nets (Figures 11 and Figure 20) where sub‐processes are represented as macro‐transitions and where grey nodes are highlighted to capture the start and end of single sub‐processes. 

Reviewer #2 concern #6

Reviewer’s comment: Considering paragraph 3: Two Ju/’hoan adhesive productions: 3) Individual sub‐processes: a) the presence of redundant situations should be better motivated. For instance, in Figure 8 the transition scales1 ‐> turn‐> scales2 may be reduced to scales1 with the elimination of the turn transition; b) once the sub‐process 7 (see Figure 9) is activated it never stops; c) the initial and final markings of each sub‐process is not always straightforward. This is mainly due to the absence of the marking in the relevant model. It makes it difficult to analyse the dynamics of the models.

Author’s response: The presence of redundant situations is motivated by the inclusion of these in the original ethnographic descriptions which, are included in section 3.1.I. For the specific example of scales1>turn>scale2 we wanted to capture the part of the description where the scales are heated on both sides.

 For the subprocess 7 of the A. Coranica, the process stops temporarily when there are no more scales on a large block and completely ends when all fleshy scales have been pounded. To explain this better, we included a text to indicate this (in red):

If activated, subprocess 7 can stop temporarily when there are no tokens in the place ‘Scales on large block’ and it 7 finishes when u scales (see subprocess 6) have gone through the pounding and folding sequence.

 We agree with the reviewer’s observation. We consider that the tables included in the newly added Appendix (S4) resolve these comments. These tables include the initial marking required to finish each subprocess and the final marking of each subprocess

Reviewer #2 concern #6

Reviewer’s comment: Considering paragraph 3: Two Ju / ’hoan adhesive productions: 4) The simulation is based on the number of people available (p = 1 to 11) but with the value at 1 of the markers involved in the sub‐process combined with the lack of an overview of the whole process, it is difficult to understand where the variations of p affect the space of reachable states; in fact the dynamics of many sub‐processes is independent of the variation of p.

Author’s response: We agree with the reviewer’s observation that the dynamics of many sub-processes are independent of the variation p. We analysed in the models and simulations the effect of the variation of p in the dynamics between subprocesses. We consider that the inclusion of figures 11 and 20, and the information in the new Appendix (S4) provides new information to clarify this approach. The internal dynamics of the subprocesses and the effects of variations of available makers for executing subprocesses is a future direction of research. For example, one can use the local variables assigned to the makers involved in a subprocess to study further the effects of the internal dynamics of the subprocess in the overall complexity of the production system. This reflection is now included in the further research section (4.4).

---

## [Decision Letter · Decision Letter 1]

15 Nov 2022

Modelling and measuring complexity of traditional and ancient technologies using Petri nets

PONE-D-22-18669R1

Dear Dr. Fajardo,

We’re pleased to inform you that your manuscript has been judged scientifically suitable for publication and will be formally accepted for publication once it meets all outstanding technical requirements.

Kind regards,

Fabrizio Pecoraro

Academic Editor

PLOS ONE

Additional Editor Comments (optional):

All the reviewers' comments and the paper can go through the next step of the editorial process.

Reviewers' comments:

Reviewer's Responses to Questions

**Comments to the Author**

1. If the authors have adequately addressed your comments raised in a previous round of review and you feel that this manuscript is now acceptable for publication, you may indicate that here to bypass the “Comments to the Author” section, enter your conflict of interest statement in the “Confidential to Editor” section, and submit your "Accept" recommendation.

Reviewer #1: All comments have been addressed

Reviewer #2: All comments have been addressed

2. Is the manuscript technically sound, and do the data support the conclusions?

Reviewer #1: Yes

Reviewer #2: (No Response)

3. Has the statistical analysis been performed appropriately and rigorously? 

Reviewer #1: N/A

Reviewer #2: (No Response)

4. Have the authors made all data underlying the findings in their manuscript fully available?

Reviewer #1: Yes

Reviewer #2: (No Response)

5. Is the manuscript presented in an intelligible fashion and written in standard English?

Reviewer #1: Yes

Reviewer #2: (No Response)

6. Review Comments to the Author

Reviewer #1: The authors have addressed my concerns, which were entirely related to their critique of the chaine operatoire method in the draft manuscript. The authors have eliminated this section of the paper altogether. I think this is a solid paper and a substantive contribution to the problem of measuring complexity in hunter-gatherer technology.

Reviewer #2: (No Response)

7. PLOS authors have the option to publish the peer review history of their article (what does this mean?). If published, this will include your full peer review and any attached files.

Reviewer #1: **Yes: **John F Hoffecker

Reviewer #2: No

---

## [Editor Report · Acceptance letter]

17 Nov 2022

PONE-D-22-18669R1 

Modelling and measuring complexity of traditional and ancient technologies using Petri nets 

Dear Dr. Fajardo:

I'm pleased to inform you that your manuscript has been deemed suitable for publication in PLOS ONE. Congratulations! Your manuscript is now with our production department. 

Kind regards, 

on behalf of

Dr. Fabrizio Pecoraro 

Academic Editor

PLOS ONE